# A micro-architectured material as a pressure vessel for green mobility

Yoon Chang Jeong ⓘ [1], Seung Chul Han ⓘ [1,2], Cheng Han Wu ⓘ [1] & Kiju Kang ⓘ [1] ✉

A shellular is a micro-architectured material, composed of a continuous smooth-curved thin shell in the form of a triply periodic minimal surface. Thanks to the unique geometry, a shellular can support external load by co-planar stresses, unlike microlattice, nanolattice, and mechanical metamaterial. That is, the shellular is the only stretching-dominated material with the highest strength at a density of less than $10^{-2}$ g/cc. Therefore, it is expected to support internal pressure, too, by the bi-axial tensile stresses like a balloon. For more than 300 years, spherical and cylindrical vessels have been viable yet compromised options for storing pressurized gases. However, emerging green mobility necessitates a safer and more spatially conformable storage solution for hydrogen than spherical and cylindrical vessels these conventional vessels. In this study, we propose to use the shellular as a pressure vessel. Due to the distinct topological nature – periodic micro-cells constituting the triply periodic minimal surface, the alternative pressure vessel can be tailored individually for spatial requirements while ensuring safety with *leak-before-break*. For a given constituent material and prescribed pressure, the achievable internal volume-per-total weight of a *P*-surfaced, cold-stretched, double-chambered shellular vessel with a number of cells more than $15 \times 15 \times 15$ can exceed the practical upper bound of both spherical and cylindrical vessels. For the applications, a thin shell with the large surface area of this micro-architecture is ideal for interfacial transfer of heat or mass between its two sub-volumes under internal pressure.

It's been more than a decade since Microlattice, Nanolattice, and Mechanical metamaterial were first introduced[1–4]. (The last one is rather used as a broad term to denote an artificial structure with mechanical properties defined by its topology rather than its composition nowadays.) They all have micro-architecture[5] (i.e., a large number of uniform cells), hierarchical structure, and ultralow densities below $10^{-2}$ g/cc with rubber-like resilience. However, despite the fascinating characteristics, the initial enthusiastic praise didn't last so long because their structures composed of extremely thin foils are too weak to be used for most practical applications. Perhaps in the real world, only one structure that can be effectively supported by such

thin foils is a balloon, namely, a pressure vessel in engineering terms, because the thin-foiled structure has only to withstand expansion, i.e., biaxial tensile stresses. Nevertheless, the three micro-architectured ultralow density materials are obviously not proper in the form as a pressure vessel. In fact, all the three materials were fabricated based on a common principle, i.e., deposition of a hard substance on a polymer template, followed by etching out the template. Thus, the hard substance formed a single continuous foil architecture, and consequently, it could play as an interface between the inner sub-volume originally occupied by the template and the outer sub-volume. If it was possible to support the pressure difference between the sub-volume, the three

[1]School of Mechanical Engineering, Chonnam National University, Gwangju 61186, Republic of Korea. [2]Present address: Reliability Research Division, Korean Construction Equipment Technology Institute, Gunsan-si, Republic of Korea. ✉e-mail: kjkang@chonnam.ac.kr

micro-architectured materials could have been used for wide applications, such as heat exchangers and so on[6,7].

The recent shift to green mobility has accelerated the need to improve the performance of pressure vessels that store and carry hydrogen[8,9]. It is crucial for next-generation pressure vessels to have less weight, high safety, and the ability to conform to irregular external spaces. However, since the Savery explosion in 1716[10], pressure vessels are known to be susceptible to failure, often by energetic explosions. High-pressure storage requires a thick-walled vessel, potentially leading to catastrophic explosive failure of the structure. To limit the severity of thick-walled vessel failure, the *leak-before-break* design philosophy as applied in the case of a thin-walled vessel has demonstrated improvements in safety[11]. Technological advancements and strict regulations have to some degree mitigated accidents involving pressure vessels, however, failures still occur[12,13]. In addition, pressure

vessels are typically designed as a sphere or cylinder, which makes them inefficient and difficult to install or carry.

Here, we propose to use the only micro-architectured material, that can support high internal pressure although being composed of an ultrathin foil, as a pressure vessel. That is a shellular[14], composed of a continuous smooth-curved thin shell in the form of a triply periodic minimal surface (TPMS)[15]. Thanks to the unique geometry, a shellular can support external load by co-planar stresses, unlike the above-mentioned three materials. That is, the shellular is the only stretching-dominated material[16] with the highest strength at a density of less than $10^{-2}$ g/cc[17]. Therefore, it was expected to support internal pressure, too, by the bi-axial tensile stresses like in a balloon, which was the initial motivation of this study. Figure 1a shows a TPMS (P-surface) as a continuous surface that sections a cube-like space into two sub-volumes bounded by the blue-shaded (outer) and the orange-shaded

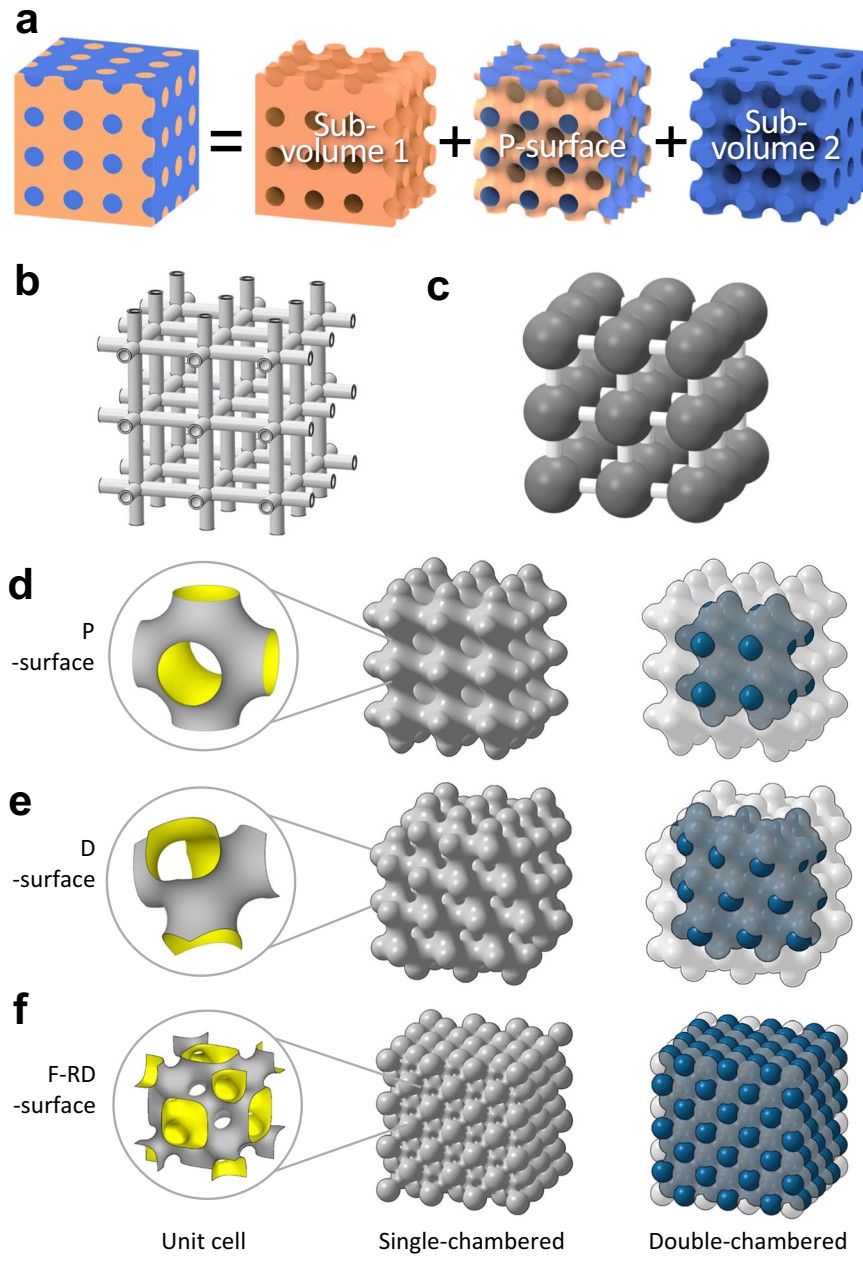

**Fig. 1 | Configurations of TPMS shellular pressure vessels. a** *P*-surface as a TPMS that sections a cube-like space into two sub-volumes bounded by the blue-shaded and the orange-shaded surfaces. Examples of high-pressure resistant structures: **b** a tubular truss and **c**, a matrix of spheres connected with tubes. Unit cells, single-chambered, and double-chambered shellular pressure vessels of three types of TPMSs, **d**, *P*- surface, **e**, *D*-surface, and **f**, *F-RD*-surface.

inner surfaces. These sub-volumes are equivalent and intertwined yet are also independent of each other. That is, the *P*-surface is the interface between the two sub-volumes, which constitutes the gas containment walls.

Our proposal is based on the results of an experiment conducted by Kolesnikova et al.[18] in 2019. They measured the critical pressures of *P*-surfaced shellulars with $3 \times 3 \times 2$ cells, composed of Ni-P, Cu, and silica, under internal pressure. The critical pressures of the most ductile shellular specimens, composed of Cu, were the highest among them, despite the low strength of Cu. Although the specimens exhibited substantial geometrical imperfections and were made of shells with thicknesses of the order of a micrometer, under internal pressure, the resistances of the Cu shellular specimens were close to that of a conventional cylindrical pressure vessel for a given *t/D* (Please note that the *D* values of shellular specimens indicated their unit cell sizes, whereas the *D* value of a conventional cylindrical pressure vessel indicates the diameter of its overall shape, and *t* indicated shell (or wall) thickness.).

Nevertheless, based on this result, one should not conclude that a shellular can be used as a pressure vessel before verifying whether its internal volume per total vessel weight is sufficiently high. For example, a truss-like structure composed of tubes and a matrix of spheres connected by tubes, as illustrated in Fig. 1b, c, respectively, cannot be used as pressure vessels, because their internal volumes are limited compared to their total weights, even if they exhibit high strength under pressure. In fact, the internal volumes per total weight of the Cu shellular specimens tested by Kolesnikova et al.[18] were approximately only half those of the cylindrical pressure vessels with identical *t/D* values, as elaborated in the Supplementary Note 1. Then, can a shellular not be a good pressure vessel? To answer this question, over the past four years, we have comprehensively studied the effects of parameters such as TPMS type, cell size, sealing caps, and sub-volumes before finally finding a route to achieve a shellular pressure vessel that outperforms conventional pressure vessels.

In this article, we summarized the findings of the aforementioned comprehensive investigation. First, finite element analysis (FEA)-based numerical simulations were performed to investigate the working characteristics of shellular pressure vessels by considering the TPMS type, cold stretching, cell size, and double chamber. As a result, we found the specific route to achieve the superior pressure vessel. In addition, shellular specimens were fabricated from copper with a TPMS (*P*-surface) and tested experimentally under internal pressure to validate and verify the pressure vessel concept more directly. Then, an attempt is made to rationalize how a shellular pressure vessel can outperform even a conventional spherical pressure vessel. Finally, the advantages and prospects of this radical design are highlighted, with a focus on practical applications.

## Results
### Selection of model
For the geometry of shellular pressure vessels, we initially considered three types of TPMSs, i.e., *P*-, *D*-, and *F-RD*- surfaces, whose unit cells are illustrated in the left column of Fig. 1d–f. The *P*-surface is well known for its high fluid permeability[19]. The shellular in *D*-surface reported high strength under compression at a typical relative density[20], and the *F-RD*-surface has a much larger surface area for a given volume than the other two[21]. Preliminary FEA structural analyses to determine the pressure resistance of the three shellulars[22] revealed somewhat uneven stress distribution over the shell. The pressure resistance is not sufficiently high to be used as a pressure vessel. Thus, to achieve a more uniform stress distribution, we slightly modified the geometry with local plastic deformation by applying an over-pressure. This is a common treatment in conventional pressure vessel fabrication, called cold stretching, used to relax residual stress, enhance strength, and verify no leaks by applying over-pressure[23].

To obtain the cold-stretched models by conducting FEA, we applied excessive internal pressure to each model until more than 85% of its surface area underwent plastic yielding. Consequently, the maximum equivalent plastic strain on the surface reached 8.6%. Subsequently, the pressure was released, and the deformed configuration was used as the cold-stretched model. In the FEA results for each of the three shellulars, although its overall shape changed just slightly, the cold-stretched model revealed more uniform stress distribution than the original model, and consequently the yield pressure at which the von Mises stress anywhere in the model first reaches the yield strength of the constituent material increased by approximately three times, as seen in Fig. S3 in Supplementary Information. The three cold-stretched shellulars were compared in terms of internal pressure resistance and the internal volume per total vessel weight, as elaborated in the Supplementary Note 2. Accordingly, as revealed by Fig. S4b in Supplementary Information, the cold-stretched *P*-shellular demonstrated the best performance, and thus was chosen for further analyses and experiments.

From the FEA results (Fig. S4a in Supplementary Information), the yield pressures of the cold-stretched *P*-shellulars normalized by the yield strength of the constituent material, $P_o/\sigma_o$, were fitted as a function of the shell thickness normalized by the cell size, *t/D*, as follows:

$$\frac{P_o}{\sigma_o} = 1.357 \times \left(\frac{t}{D}\right)^{0.9358} \text{ for } P - \text{shellular,} \tag{1}$$

### Efficiency of pressure vessel
The internal volume per total vessel weight as well as the yield pressure should be considered at a given pressure. To evaluate the vessel's efficiency in containing pressurized gas considering the internal volume per total vessel weight, we defined a new parameter termed efficiency of pressure vessel (*EPV*) as follows.

$$EPV \equiv \frac{P_o V_{in}}{\sigma_o V_s} = \frac{P_o f V}{\sigma_o A t} \tag{2}$$

Here, *V*, $V_{in}$, $V_s$, and *A* denote the overall, internal, solid constituent material's volumes, and shell's area, respectively. And *f* denotes the volume fraction, defined as the ratio of one sub-volume to the overall volume. In our TPMS shellular pressure vessel design, we used a constant volume fraction of *f* = 0.5, meaning that the two sub-volumes were identical in a unit cell.

According to Boyle's law, pressure multiplied by the internal volume of the contained gas is constant as a measure of its internal energy at a given temperature. Thus, *EPV* represents the maximum permissible internal energy of contained gas for a given strength and volume of the solid constituent material of the vessel.

### Cell size effect
Until now, we have assumed that the shellular is composed only of a shell in a TPMS. However, in reality, a shellular needs additional sealing caps on its outer openings to confine the gas as a pressure vessel. Therefore, we designed the shellulars to have hemispherical sealing caps, as shown in the middle column of Fig. 1.

According to Eq. (1), the relative yield pressure of a TPMS shellular is a function of the relative thickness, *t/D*. That is, if the cell size, *D*, is designed smaller, the thinner shell, i.e., the smaller *t* can be used to hold gas at a given internal pressure. Therefore, a pressure vessel can be created using a thin foil with a form of TPMS comprising a large number of small cells. To examine the feasibility, we performed another series of FEA. The tops in Fig. 2a–c show the *P*-shellular models with single, $3 \times 3 \times 3$, and $9 \times 9 \times 9$ cells. The cell size, *D*, of each model was determined depending on the number of cells such that its

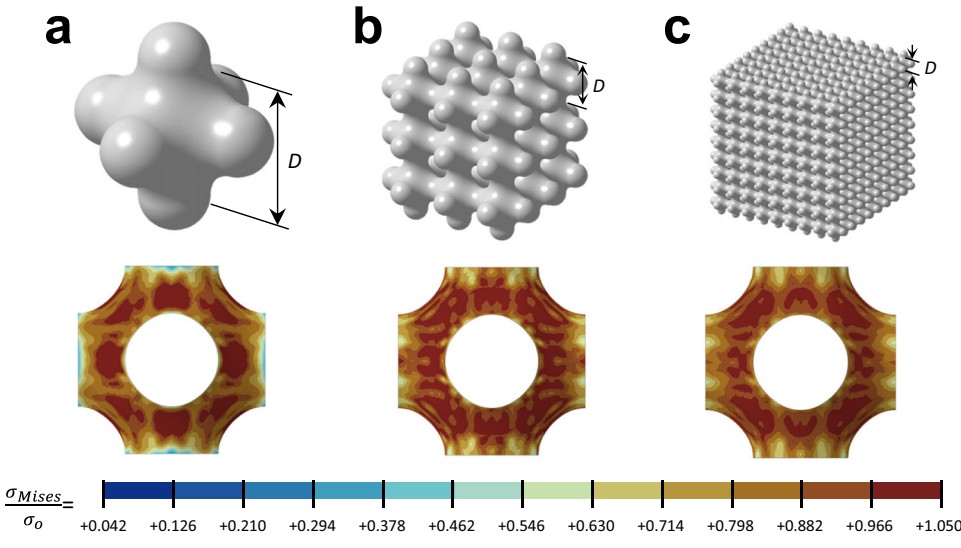

| (unit: mm) | **a** (single cell) | **b** (3x3x3 cells) | **c** (9x9x9 cell) |
|---|---|---|---|
| Cell size, **D** | 909 | 325 | 111 |
| Shell thickness, **t** | 5 | 1.8 | 0.6 |

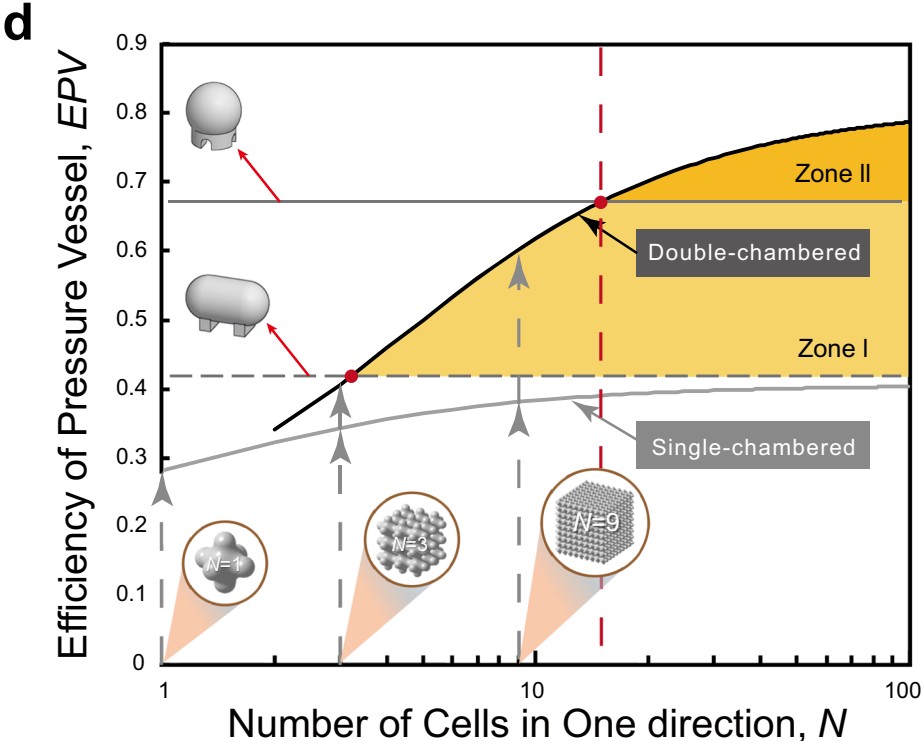

**Fig. 2 | Cell size effect of *P*-shellular pressure vessels with a constant internal volume.** Models (upper) and von Mises stress distribution in unit cells (lower) of *P*-shellular with **a**, a single, **b**, 3 × 3 × 3, and **c**, 9 × 9 × 9 cells. **d** Variation of *EPV*s with one-directional number of cells, *N*, estimated using Eqs. (3) and (4) for single- and double-chambered *P*-shellular pressure vessels, respectively.

internal volume was the same as that of a reference spherical pressure vessel with a 1 m diameter. The shell thickness, $t$, was then determined as the relative thickness was fixed as $t/D = 0.0055$, calculated for a constant yield pressure of $P_o = 0.01\sigma_o$ according to Eq. (1) and Fig. S4a in Supplementary Information. The middles of Fig. 2a–c show the von Mises stress distributed in single unit cells at the center of each of the three models. Notably, the stress distributions were almost identical as expected despite the difference in cell size and shell thickness. This

result demonstrates that a pressure vessel can be created using a continuous thin foil with a form of TPMS composed of a large number of tiny cells, having the same pressure resistance as that of a single-cell conventional pressure vessel with thick walls.

Table S2 in Supplementary Information lists the cell sizes, shell thicknesses, surface areas, weights, and *EPV*s of *P*-shellular pressure vessels with single, 3 × 3 × 3, 9 × 9 × 9, and 100 × 100 × 100 (i.e., one million) cells in comparison with those of the conventional cylindrical

and spherical pressure vessels. See the Supplementary Note 3 for the derivation of $EPV$s for conventional vessels and $P$-shellular vessels. It was observed that the $EPV$s of the conventional cylindrical and spherical vessels were constant at $EPV = 5/12$ (=0.417) and $EPV = 2/3$, respectively, regardless of the applied pressure or the yield strength of constituent material.

Thus, the general solutions of cell size, surface area, and shell thickness for $N \times N \times N$ cells can be derived for a fixed internal volume, $V_{in}$, and yield pressure, $P_o = 0.01\,\sigma_o$. Accordingly, $EPV$ can be expressed as follows.

$$EPV = 0.475 \times \frac{(8N + \pi)}{(9.36N + 3\pi)} \tag{3}$$

The $EPV$ increases with the number of cells in the $P$-shellular vessel. For example, in the case of one million ($100 \times 100 \times 100$) cells, a $P$-shellular vessel could be fabricated using a foil with a thickness of $t = 54\,\mu m$ and a cell size of $D = 10\,mm$ for the given internal volume. And its efficiency would be $EPV = 0.39$, which is 94% and 59% of those for the conventional cylindrical and spherical pressure vessels, respectively. That is, the single-chambered $P$-shellular vessel cannot outperform even the conventional cylindrical pressure vessel.

### Double-chambered vessel

A TPMS shellular has two sub-volumes, as shown in Fig. 1. Hence, the two sub-volumes can be individually and simultaneously used as chambers holding a pressurized gas. The right column of Fig. 1d–f depict the double-chambered versions of the single-chambered shellular pressure vessels with the $P$-, $D$-, and $F$-$RD$- surfaces, shown in the middle column of Fig. 1. In each shellular pressure vessel, the exterior first chamber of one sub-volume is depicted to be semi-transparent to reveal the interior second chamber of the other sub-volume with additional sealing caps on its outer openings. Since the second chamber can be formed by extra sealing at the necks between the cells of the first chamber, the second sub-volume likely has a smaller number of cells. For a double-chambered $P$-shellular pressure vessel with $N \times N \times N$ cells, the general solutions of the internal volume, $V_{in}$, and surface area, $A$, can be derived for a given yield pressure of $P_o = 0.01\,\sigma_o$. Accordingly, $EPV$ can be expressed as follows.

$$EPV = 0.475 \times \frac{8N\left(1 + \left(\frac{N-1}{N}\right)^3\right) + \pi\left(1 + \left(\frac{N-1}{N}\right)^2\right)}{9.36N + 3\pi\left(1 + \left(\frac{N-1}{N}\right)^2\right)} \tag{4}$$

Figure 2d shows the $EPV$s estimated according to Eqs. (3) and (4) for the single-chambered and double-chambered $P$-shellulars, respectively. The $EPV$ for the single-chambered $P$-shellular vessel had the upper limit, $EPV = 0.406$. Thus, the $EPV$ cannot be higher than that of the conventional cylindrical pressure vessel, $EPV = 0.417$. In contrast, the $EPV$ for the double-chambered $P$-shellular vessel rapidly increased with $N$. At $N = 4$, the $EPV$ was already higher than that of the conventional cylindrical pressure vessel, and at $N = 14$, the $EPV$ reached that of the conventional spherical pressure vessel. Accordingly, in the case of one million cells ($N = 100$), the $EPV$ of the double-chambered $P$-shellular vessel with $t = 54\,\mu m$ and $D = 10\,mm$ was estimated to be $EPV = 0.79$, which is 188% and 119% of the values for the conventional cylindrical and spherical pressure vessels, respectively. In Fig. 2d, Zone I and II represent the domains where the double-chambered $P$-shellulars outperformed the conventional cylindrical and spherical vessels, respectively, in the $EPV$s.

The seventh subsection of the Supplementary Note 2 shows that the yield pressure obtained when both sub-volumes are pressurized is the same as that obtained when only one sub-volume is pressurized.

### Measurement of yield pressures

The $P$-shellular specimens were prepared and tested under internal pressure for the direct validation of feasibility. The technical details are provided in the Supplementary Note 6, 7. The experiments aimed to check the validity of only Eq. (1) for the yield pressure. The other equations are just logical consequences based on the equation. The specimens were designed with miniature overall size and fine cells for easy fabrication in our laboratory and to check the scalability of the concept. The yield pressure of the shellular specimens was measured under internal pressure using a small-scale test method[18]. The overall test setup and a close-up of the double-chambered specimen with a small tubular needle inserted are shown in Fig. 3a, b, respectively.

Figure 3c shows the load-displacement curves measured for the two representative single-chambered specimens with $t/D = 0.00877$ and 0.00079, tested under internal pressure. In each specimen, the two curves before and after cold stretching, which are denoted in blue and black, respectively, overlapped to identify its effect. The difference in their shell thicknesses was more than ten times. Consequently, the load levels around and after the initial yield differed by a factor of more than ten. Nevertheless, both specimens exhibited common features. Initially, they revealed non-linear lower slopes and continuous changes with unclear yield points in the curves. After the initial yielding, the curves were almost flat, that is, the load levels were kept constant for a while, indicating the specimens underwent large-scale plastic deformation, specifically under cold stretching. However, in the second loading (after cold-stretched), they revealed the linear higher slopes and sudden changes with clear yield points, demonstrating the significant effect of cold stretching.

Figure 3d shows the load-displacement curves measured for the two representative double-chambered specimens with $t/D = 0.0107$ and 0.00113. Despite the existence of the second sub-volume, their load-displacement curves revealed similar behaviors to those of the single-chambered specimens. In addition, because the two sub-volumes are connected to each other via the holes made during specimen preparation, the interior shell is not subjected to any stress. Thus, the specimen is free from any defects in the interior shell, which would be the double-chambered design's practical advantage in addition to the higher $EPV$.

For each specimen, the load at the yield point was divided by the cross-sectional area of the syringe plunger, as shown in Fig. 3a, to obtain the yield pressure. Figure 3e summarizes the results for the single- and double-chambered specimens in the left and right panels, respectively. Here, the yield pressures and the shell thicknesses are normalized by the yield strength of the Cu foil, $\sigma_o$, and the unit cell size, $D$, respectively. For comparison, the estimations by elementary mechanics (Eqs. (S2) and (S3) in Supplementary Information) for conventional pressure vessels are plotted as the solid black and dashed gray lines, respectively. Furthermore, the estimations by Eq. (1), derived from the FEA for the unit cell of cold-stretched $P$-shellular, are also plotted as the solid violet lines.

Three points are noticeable in Fig. 3e. Firstly, the relative thickness ranged from $t/D = 7.9 \times 10^{-4}$ to $2.2 \times 10^{-2}$, corresponding to the shell thicknesses of $t = 4$ to $110\,\mu m$ with a constant cell size of $D = 5\,mm$. While the upper limit of $t$ was set by the maximum pressure allowable for the syringe and needle of the loading system, the lower limit was attributed to the fact that the electroless plating left the roughness on the surface, scaled by the maximum peak-to-valley value of $4\,\mu m$ or less even for the thickness around the lower limit, as shown in Supplementary Note 10. Consequently, data from the specimens with $t/D < 8 \times 10^{-4}$, corresponding to the shell thickness of $t = 4\,\mu m$, couldn't be obtained.

Second, the upper bound of the measured yield pressure data agreed well with the estimations made by Eq. (1), derived from the FEA for the cold-stretched $P$-shellular, and all the data near the upper bound were obtained from the cold-stretched specimens. Note that

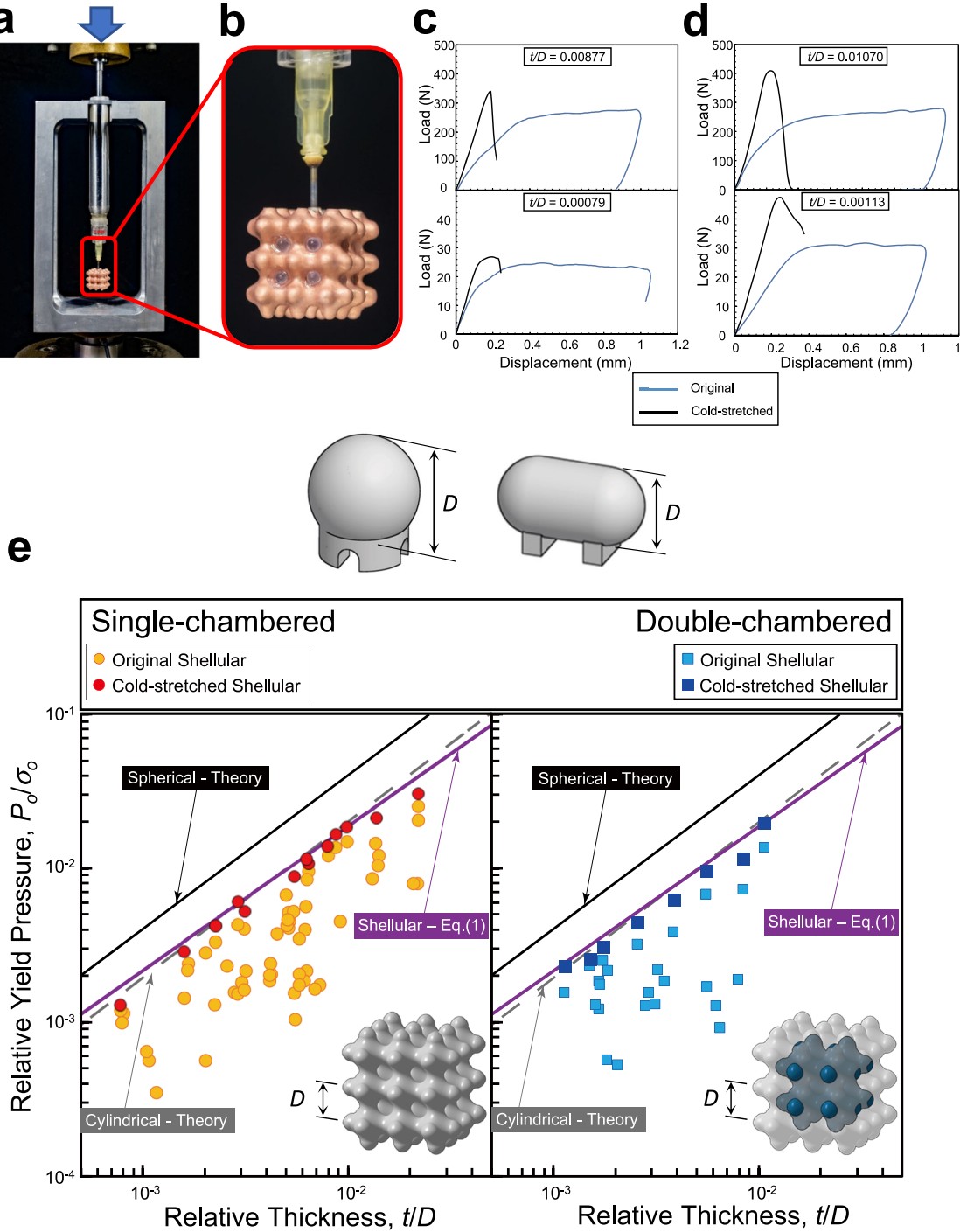

**Fig. 3 | Experimental results. a** Overall test setup, **b** close-up of the double-chambered specimen mounted with a small tubular needle connected. **c, d** Load-displacement curves at two relative thicknesses, $t/D$, measured from the single- and double-chambered specimens, respectively, under internal pressure, **e** data of relative yield pressure versus relative thickness measured for $P$-shellular specimens in comparison with the theory of conventional pressure vessels.

the experimental results comprise those measured from many specimens that failed early during cold stretching, leading to low yield pressures, which was also attributed to the rough and non-uniform thickness of the shell. Therefore, the yield pressures of the specimens that were cold stretched without any problems were consistent with the estimations made by Eq. (1).

Finally, the double-chambered specimens revealed a trend very similar to that of the single-chambered specimens in the yield pressure versus the relative thickness data. Thus, a double-chambered shellular

vessel can be designed based on the yield pressure evaluated for its single-chambered counterpart.

These experimental results validated the numerical simulation FEA results fitted by Eq. (1) for the single and double-chambered specimens. Thus, we believe that the *EPV*s, which is a measure of the pressure vessel's performance, estimated by Eqs. (3) and (4), were correct. Consequently, the superiority of the shellular pressure vessels over the conventional spherical and cylindrical vessels was proved.

## Discussions

### Feasibility of shellular pressure vessels being superior to conventional spherical vessels

Although the logic applied to estimate the *EPV* of the shellular vessels, illustrated in Fig. 2d, is obvious and sound, and it has been demonstrated that the *EPV* value can be higher than that of a spherical vessel, one may still distrust that a shellular pressure vessel can outperform a spherical vessel. This is because the spherical vessel is known as the only structure that supports internal pressures due to uniform biaxial coplanar stresses, regardless of the location and direction over the entire surface, and can, therefore, support internal pressure in the most efficient manner with minimal weight. However, it should be noted that the shellular vessel could be superior to the conventional spherical vessel only in cases that it uses double chambers and a large number of cells. For a double-chambered shellular vessel, one portion of the shell, a face of which is exposed to the outer atmosphere while the other face is in contact with the interior pressurized gas, is defined as the outer shell. Hemispherical sealing caps are included in the outer shell. The other portion of the shell, both faces of which are in contact with the interior pressurized gas, is defined as the interior frame. Figure 4a illustrates the outer shell and interior frame, which are marked with two different colors, of a double-chambered shellular vessel (By contrast, for a single-chambered shellular vessel, the entire shell area serves as the outer shell.).

Figure 4b depicts the variation in the area and thickness of the outer shell and interior frame with the number of cells *N* along one direction under a constant yield pressure $P_o = 0.01\,\sigma_o$, as assumed in Fig. 2, and total internal volume equal to that of a spherical vessel with an inner diameter of 1 m. The surface area of the outer shell remains almost constant regardless of *N*, whereas the area of the inner frame increases in proportion to *N* and soon considerably exceeds that of the outer shell. Consequently, the sum of the two areas increases linearly with *N*, that is, as the cell size *D* decreases. By contrast, the shell thickness *t* decreases rapidly at *N* values lower than 20 and decreases more slowly thereafter at higher *N* values. As a result, as depicted in Fig. 4c, the solid volume of the outer shell, which is calculated by multiplying its surface area with its thickness *t*, decreases rapidly from the initial value ($V_s/V_{in} = 0.027$ at $N = 2$) and soon converges to 1/17 of the initial value, whereas the solid volume of the interior shell increases rapidly from the initial value ($V_s/V_{in} = 0.00225$ at $N = 2$) and soon converges to five times the initial value. Consequently, the sum of the two solid volumes (i.e., total vessel weight divided by the density of the constituent material) decreases from the initial value ($V_s/V_{in} = 0.0293$ at $N = 2$), drops below that of the spherical pressure vessel ($V_s/V_{in} = 0.0149$) at $N = 15$, and finally converges $V_s/V_{in} = 0.0133$ at $N = 40$. Considering the variation of shell thickness, depicted in Fig. 4c, the total weight of the double-chambered shellular vessel, which is lower than that of the spherical vessel, can be attributed not only to the fact that the second chamber is secured by simply adding the minimal weight of the sealing caps to the single-chambered version but also the fact that the shell thickness *t* rapidly decreases as the number of cells *N* increases. Here, the solid volume is expressed as a dimensionless form, solid ratio by dividing with the constant internal volume. The solid volumes are derived in the Supplementary Note 4.

Another qualitative explanation is presented in Fig. 4d, e. If a single sphere of diameter $D_o$ is divided into an $N \times N \times N$ matrix composed of small spheres of diameter $D_o/N$, the internal volume of the matrix remains constant, but the surface area *A* increases in proportion to *N*. However, if the pressure resistance is constrained to be constant, the shell thickness *t*, that is, the shell diameter *D*, should decrease in proportion to *N*. Consequently, the solid volume $A \times t$ remains constant regardless of *N* or the shell diameter. This is described in the Supplementary Note 4. Figure 4d presents an image of the single sphere and those of the $3 \times 3 \times 3$ and $9 \times 9 \times 9$ matrices of the small spheres at a constant scale. All of them have identical pressure

resistances and internal volumes. Interestingly, the overall size in the perspective views (i.e., the diagonal dimension) increases as the number of spheres in a row *N* increases, although the width, depth, and height remain constant. By contrast, according to Fig. 4e, the overall size of the double-chambered shellular in the perspective view decreases. This is because the proportion of the second sub-volume increases with *N* when the internal volume is constant. For example, a double-chambered shellular comprising $3 \times 3 \times 3$ (=27) cells has a second volume of $2 \times 2 \times 2$ (=8) cells, as depicted in Fig. 1d, which occupies a small portion (=8/(8 + 27) ≈ 0.23) of the entire internal volume, even if the volumes of the sealing caps are ignored. By contrast, a shellular comprising $9 \times 9 \times 9$ cells has a second volume of $8 \times 8 \times 8$ cells, which occupies a considerably larger proportion of the internal volume (=$8^3/(8^3 + 9^3)$ ≈ 0.41). Because the proportion of the second sub-volume increases with *N*, the overall size scaled by the sum of the first and second sub-volumes decreases for a constant internal volume. Consequently, the cell size *D* and shell thickness *t* become smaller than that obtained by simply dividing the initial overall size with *N*, which causes the solid volume of the double-chambered shellular to decrease with *N*, as illustrated in Fig. 4c.

### Technical difficulty associated with its fabrication

Another challenge that must be solved to facilitate usage of the shellular pressure vessel in practical scenarios is the technical difficulty associated with its fabrication. Owing to the complex geometry involved, it is challenging to fabricate a shellular vessel, especially one composed of a large number of cells. However, it was not excessively difficult to fabricate the miniature specimens described herein thanks to the high permeability (for the electroless plating solution) of the smooth curved minimal surface, particularly in cases of the double-chambered specimens. Actually, 13 (19%) specimens out of 68 single-chambered specimens tested herein exhibited successful cold-stretching and then sufficiently high-yield pressures, whereas 8 (25%) out of 32 double-chambered specimens exhibited successful behaviors. Considering that even a single defect is not allowed in pressure-supporting vessels, the success ratios achieved by the shellular specimens with shell thicknesses of $t = 4$–110 μm ($t/D = 0.00079$–0.022) were considered excellent. Particularly, these results demonstrated that the double-chambered specimens were less sensitive to defects, which indicated that a certain quantity of defects could be allowed in the interior frame, unlike the outer shell. In fact, because both sub-volumes of the double-chambered models supported the same pressure, there was no pressure difference between the two faces of the interfacial shell, which acted as the interior frame. Consequently, the stresses in the interior frame were lower than those in the outer shell, as marked in brown near the inner area and in red near the outer area in Fig. S5c in the Supplementary Information.

Then, a question arises. What is the role of the interior frame in a double-chambered shellular vessel? To answer this question, we conducted a new series of FEAs to investigate the variation of the Mises stress distributed in double-chambered shellular pressure vessel models with $9 \times 9 \times 9$ cells under internal pressure of $P_o = 0.002\,\sigma_o$, as the layers of the interior frame are removed one by one from the center. Figure 5a compares the results of three models with intact interior frames, one cell at the center removed, and three layers removed. These results indicate that the removal of even a single cell located at the center causes an apparent change in the stress state near the center and the removal of the three layers causes a substantial change in the stress state even on the outer shell. Another FEA was conducted to investigate the effects of defects existing on the interior frame. Figure 5b compares Mises stresses distributed in the $3 \times 3 \times 3$ cell models under internal pressure of $P_o = 0.002\,\sigma_o$ with the intact interior frame and with the elements in 5.30% of its total area missed, simulating defects such as holes, as shown in the middle between the two models. Although the missed elements apparently caused the

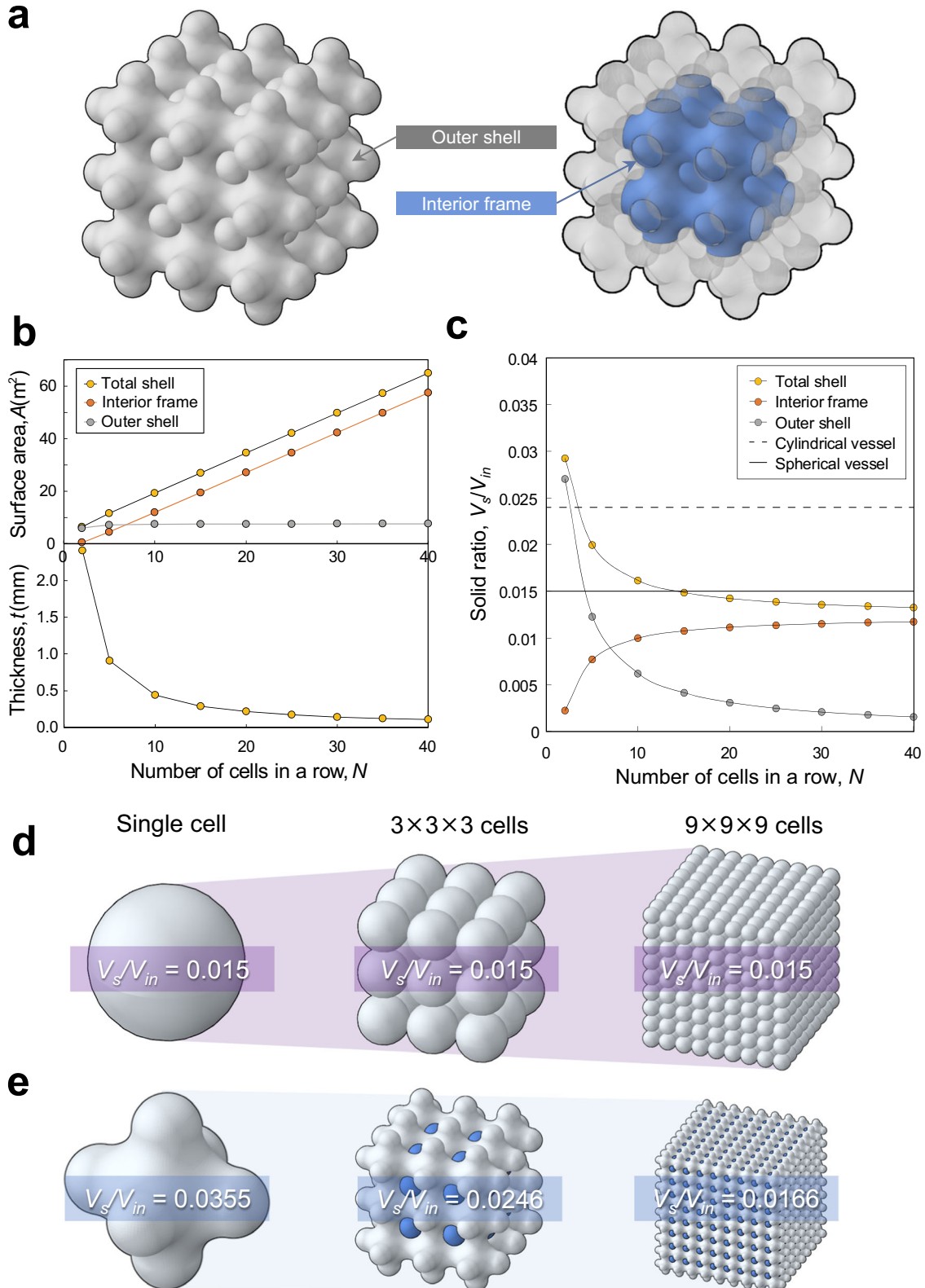

**Fig. 4 | Feasibility of shellular vessel being superior to conventional spherical vessel. a** Double-chambered shellular vessel with its outer shell and interior frame marked in two different colors. Variations in (**b**) surface area and thickness and (**c**) solid volumes of the outer shell and interior frame with the number of cells $N$ along one direction under a constant yield pressure $P_o = 0.01\ \sigma_o$ and fixed total internal volume. **d** A single sphere of diameter $D_o$, and $3 \times 3 \times 3$ and $9 \times 9 \times 9$ matrices composed of small spheres having diameters of $D_o/3$ and $D_o/9$, respectively. **e** Single shellular and double-chambered shellular vessels with $3 \times 3 \times 3$ and $9 \times 9 \times 9$ cells with identical internal volumes and pressure resistances.

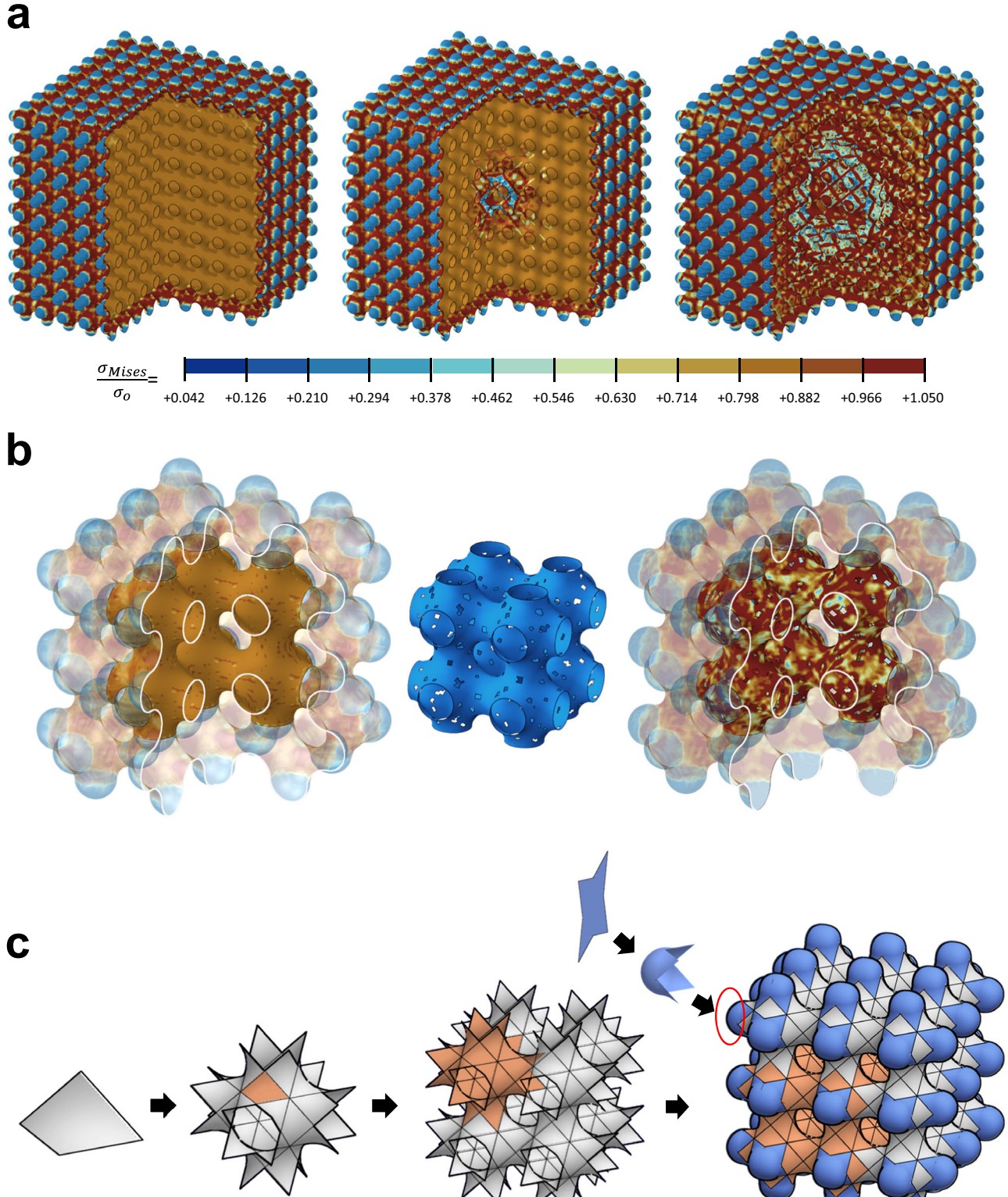

**Fig. 5 | Technical difficulty in fabrication of shellular pressure vessel. a** Variation of Mises stress distributed in double-chambered shellular pressure vessel models with $9 \times 9 \times 9$ cells under a constant internal pressure $P_o = 0.002\ \sigma_o$ with an intact interior frame, one cell at the center removed, and three layers removed. **b** Mises stresses distribution in the $3 \times 3 \times 3$ cell double-chambered models under an internal pressure $P_o = 0.002\ \sigma_o$ with an intact interior frame and with elements in 5.30% of its total area missing to simulate defects such as holes, as shown in the middle. **c** *P*-surface, composed of many regular quadrilaterals in an anti-clastic curve, implying that a large shellular pressure vessel can be fabricated by bending and welding of tiny pieces of thin shells.

stress concentration near them, the stresses distributed on the outer shell didn't exhibit any change. In Fig. 5b, the areas revealing the interior frames are highlighted to identify it from the outer shells, which are a little transparentized and darkened.

In summary, the interior frame plays an important role in holding the outer shell in place, as illustrated in Fig. 5a. Nevertheless, a large number of defects can exist on this frame without deteriorating the pressure resistance of the entire shellular vessel, as depicted in Fig. 5b. This was also validated by the high success ratios of double-chambered specimens over their single-chambered counterparts, as mentioned above. The above logic holds, regardless of the scale of shellular vessels. Because a TPMS is composed of many constant quadrilaterals in an anti-clastic curvature[24], thin shells are cut and bent into a regular quadrilateral shape and then welded to each other to build a large-scale shellular pressure vessel, as illustrated in Fig. 5c. In this case, although the thin shells themselves are easy to weld to each other by using a focused energy source, such as a fiber laser[25], the process of quality welding along the complicated contours in three-dimensional space would be technically challenging. Nevertheless, because defects are allowed in the interior frame, as mentioned above, it should be feasible to manufacture large-scale shellulars by employing a robot-based rough technology to build the interior frame, while employing a more precise technology to guarantee defect-free welding for building the outer shell, as in the fabrication of a conventional pressure vessel.

## Concluding remarks

The results of the structural simulations and experiments presented so far verified the feasibility of the concept of a pressure vessel comprising a large number of small cells made of a thin, continuous, and smooth shell. In fact, the relative thicknesses in a range of approximately $0.001 < t/D < 0.01$, considered in this work, cover most thin-walled pressure vessels for daily use and the industrial sector, from aluminum beer cans to propane gas tanks. Therefore, the findings of this study can be applied to most thin-walled pressure vessels regardless of the scale. The radical design has several advantages over conventional pressure vessels as follows:

Freedom of the overall shape design. Since it is composed of a large number of tiny shellular cells, one can design a pressure vessel with an arbitrary overall shape by changing the arrangement pattern of the cells, similar to Lego® blocks.

Realization of never-bursting high-pressure vessels. A conventional pressure vessel should be fabricated with a thicker wall to resist higher pressure. However, the thick wall may cause a catastrophic burst because a crack progressively grows in the wall to finally rupture before the crack penetrates through the wall and then lets the gas leak to release the internal pressure. In contrast, a shellular pressure vessel with numerous cells is fabricated with a thin continuous shell, which guarantees safety by means of leak-before-break[11].

Potential use of 2D materials for reinforcement. A shellular is composed of a continuous smooth thin shell. Therefore, the surface is favorable for conformal deposition of an ultra-strong 2D material, such as graphene with a breaking strength of 130.5 GPa[26], boron nitride (BN) with breaking strength of 70.5 GPa[27], and transition metal dichalcogenides (MoS$_2$ with a breaking strength of 27 GPa)[28,29]. Hence, a flexible, safe, and ultra-high-pressure vessel can be realized.

Multi-functionality. A thin shell with the large surface area of a shellular is good for interfacial transfer between its two sub-volumes. Heat, ions, and mass can be transferred through the metallic shell, catalyst-coated electrolyte membrane, and semi-permeable membrane to function as a heat exchanger, proton-exchange membrane fuel cell[6], and tissue engineering scaffold[7], respectively. Actually, not only the heat exchanger, such as a boiler but also the fuel cell and scaffold need to resist internal pressure[30,31]. As illustrated in Fig. 2d, even if half of the interior space is used for an additional function, the *EPV* could come close to that of a cylindrical vessel.

## Methods

### Establishment of finite element models

Finite element models of shellulars were prepared using the Surface Evolver software and re-meshed with quadratic shell elements (S4) through HyperMesh® (Altair Engineering, Inc., MI, USA). The parameters were set with a fixed cell size of 5 mm and element size of ~70 μm, while the shell thickness varied within the range from 0.5 to 50 μm.

To establish the cold-stretched models, an initial step involved subjecting the structures to excessive internal pressure under periodic boundary conditions. Subsequently, the applied pressure was released, and the residual stress resulting from the deformation was systematically eliminated. This process was crucial in establishing accurate representations of the cold-stretched configurations for our analyses and evaluations.

### Finite element analyses

Nonlinear simulations were conducted to assess the pressure resistance of the shellular structures using the standard solver of the commercial software Abaqus. The Young's modulus, Poisson's ratio, and yield strength were given as $E = 100$ GPa, $v = 0.3$, and $\sigma_o = 120$ MPa, respectively. Perfectly plastic behavior with no hardening was used for the analysis. To ensure a precise estimation of pressure resistance, the applied pressure was increased in steps of 1% of the expected yield pressure during the simulations.

### Fabrication of specimens

The *P*-shellular specimens were prepared with a cell size of 5 mm, shell thickness of 4–110 μm, and 3 × 3 × 3 cells. First, a negative template of a water-soluble polymer, polyvinyl alcohol (PVA), was prepared using a 3D printer (Ultimaker S3, Ultimaker B.V., Netherlands). Subsequently, a poly-methyl methacrylate (PMMA) template was created by filling cold-curing resin (CCR, Technovit® 5071 and universal liquid, Heraeus Kulzer GmbH, Germany) into the PVA template as a mold, and the PVA mold was then etched out. Next, the template underwent a chemical treatment, including the Han's treatment[17], to provide smooth silhouette of TPMS and sufficient surface roughness needed for electroless plating.

The template was subjected to electroless plating to deposit a metallic copper layer, which becomes the outer shell of the shellular pressure vessel specimen. The process involved dipping the template in an aqueous solution of HCl for 3 min, followed by immersion in another aqueous solution of PT-Activator (Young-In Plachem Co. Ltd., Korea) and HCl for 5 min at 40 °C for activator coating. Subsequently, the template was immersed in a 10% NaOH aqueous solution (accelerator) for 5 min to remove tin from the surface. The final step involved copper plating on the surface using a commercial process based on ELC-250 solution (Young-In Plachem Co. Ltd., Korea) at 70°C and pH levels of 11.8 - 12.1. A shellular specimen was obtained by etching out the PMMA interior through small holes, exposed by polishing the top face. Etching was performed by dipping in THF for 2 days. Finally, for sufficient ductility, all shellular specimens were annealed at 260 °C for 3 h in an electric oven with an argon atmosphere. Please refer to the Supplementary Note 6 for technical details.

### Sealing

For the hermetical sealing of the shellular specimen, the inner space of the top layer with the holes was filled with polydimethylsiloxane (PDMS), while 1.1 ml of a 60% glycerol aqueous solution filled the void underneath the PDMS sealing. SYLGARD™ 184 (Dow Chemical Co.) was used for the PDMS sealing. Please refer to the Supplementary Note 6 for further details.

### Preparation of double-chambered specimens

A double-chambered specimen was prepared by plugging the openings on the outer faces of a single-chambered specimen using

polystyrene beads (2 mm diameter) coated with PDMS after making holes on the interior shell for simultaneous pressurization of both sub-volumes during the internal pressure tests.

### Preparation of samples for tensile test and surface roughness measurement

We prepared a PMMA bar with a circular cross-section of 6.37 mm diameter. Surface treatment, electroless plating, and heat treatment were conducted in the same way as those performed for the shellular specimens. Finally, the specimens were cut into a flat coupon shape with dimensions of 3 mm × 20 mm.

### Measurement of yield pressures

Hydraulic pressure was applied to the interior of specimens using an electro-hydraulic material test system, INSTRON 8872, along with a specially designed frame and a stainless-steel syringe. The pressure was applied by pushing the syringe's plunger at a constant displacement rate of 0.01 mm/s. The plunger's displacement was measured using a linear variable displacement transducer (LVDT) built into the test system. To ensure precise measurements, two additional load cells with different capacities were employed to measure the applied force. Please refer to the Supplementary Note 7 for technical details.

### Tensile tests of Cu foils

The tensile properties were evaluated using a specially built tensile test machine and stainless-steel coupons that were used by Han[17]. The tensile test speed was 0.01 mm/s. Strain was measured by counting the pixels on the series of digital images, which were taken for the gauge section during each tensile test. Please refer to the Supplementary Note 8 for technical details.

### Surface roughness of Cu shells

The surface profiles of electroless plated copper were measured using a white light scanning interferometer (NV-E1000, NanoSystem Co., Ltd, South Korea). The surface profiles were treated by a function of MATLAB®, to remove long-term trends for emphasizing short-term changes.

### Data availability

The data for Figs. 2d, 3c–e, 4b, c, Figs. S1e, S4, S9b–d, S10b, d, f, h are provided as a Source Data file. The raw data and load-displacement curves measured from the internal pressure tests for all specimens are provided in Supplementary Data 1. Source data are provided with this paper.

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

## Acknowledgements

We wish to express our deepest gratitude to Prof. Tongbeum Kim and Dr. Moxiao Li from Nanjing University of Aeronautics and Astronautics for

their valuable advice and help. We also thank Dr. Michael D. Atkins from the University of the Witwatersrand for his technical advice. K.K. This work was supported by the National Research Foundation of Korea (NRF) grant funded by the Korean government (MSIT) (No. 2021R1A2C3007705).

## Author contributions

K.K. conceptualized and supervised the project. Y.C.J. and S.C.H. designed the fabrication process. Y.C.J. prepared specimens, performed the experiments, and analyzed the data. Y.C.J., S.C.H., and W.C.H. developed the pressure vessel model and performed numerical simulations. K.K. prepared the manuscript with input from all authors. K.K. and Y.C.J. prepared all figures and reviewed the manuscript.

## Competing interests

The authors declare the following competing interests. Three patent applications have been filed by the Chonnam National University, Republic of Korea on THREE-DIMENSIONAL SHELL STRUCTURE, PRESSURE VESSEL HAVING SAME, AND MANUFACTURING METHOD THEREFOR: Korea patent application (10-2018-0041156, approved), PCT (PCT/KR2019/000953, filed), and Chinese patent application (2019800248991, filed), on which K.K., Y.C.J., and W.C.H. are listed as inventors. The remaining authors declare no competing interests.
