## [Peer Review File · Nature Communications]

A Micro-Architected Material as a Pressure Vessel for Green MobilityReviewers' comments:

Reviewer #1 (Remarks to the Author):

The submitted manuscript deals with the novel design of pressure vessels called by the authors a shellular. In the 'Experimental results' section they describe the P-shellular type, although in previous sections there were described different types. Moreover, similar study was published by the authors in: Failure of P-surfaced Shellular subjected to internal pressure (<https://doi.org/10.1063/1.5066578>). I consider this as the biggest weakness of the manuscript and it should be explained in the first place, what is the main novelty comparing to that work. Moreover, the following issues should be addressed in the major revision:

1. Did you perform the analysis of mesh size element influence on the results?
2. What was the level of plastic deformation occurring in the material after cold-stretching?
3. How did you perform the strain measurements? Did you use Digital image correlation (DIC) method or just measuring the pixel displacements from regular photos?
4. How would you explain the different Young Modulus for each type of foil? (as it can be observed in Fig. S7B). It may be noticed that the highest stiffness was obtained for 27.5 μm foils? Do you have explanation for that?
5. What type of machine was used in the mechanical tests with what range of force measurements? Was it proper for such a 'delicate' specimens?
6. The abstract should include some quantitative results regarding your research.

Reviewer #2 (Remarks to the Author):

The paper is interesting and presents a promising technology that may improve pressure vessel technology

the following comments may improve clarity of presentation and understanding

the paper deals with triply periodic minimal surfaces (TPMS). What does this mean? why are they called that?

Equation 1. Parameter f was not defined

table s1 shows cell sizes, shell thicknesses, surface areas, weights, and EPVs. This seems like important information. Should this be included in the main article?

line 167 claims that the vessel efficiency is higher than that of a spherical vessel. Is this possible? As I remember, a sphere has a uniform stress distribution where every location has equal stresses in both directions along the surface. How can anything be better than that?

In a vessel like this made of individual components I believe sealing all the joints may be a challenging problem. How is this handled for a "real" vessel? if the individual p-surfaces need to be joined to individual outer membranes, how do I make sure that the joints will not leak? This should be explained in the main text.

line 203 mentions equations 6 and 7 but these are not included in the text

the authors mention hydrogen storage as a potential application. However, I question whether a metal-based system like this (even when covered with graphene) would be excessively heavy for automotive applications?

Lines 246-247: the authors report "tensile strength" for graphene, "fracture strength" for boron nitride, and "strength" for transition metal dichalcogenides. Do these three terms refer to the same thing? If they do, the authors should use the same term (tensile strength?) for all. Otherwise, why do they report different parameters for the three?

Reviewer #3 (Remarks to the Author):

This paper presents a numerical and experimental study of shellular materials as a pressure vessel. The authors have published a number of papers on shellular material systems, as a form of TPMS which have been studied for a wide range of applications and also found in many biological systems. This study employs the finite element method (using Abaqus) to conduct the stress analyses, apply some plastic deformation based on the yield stress of the material (cold-stretched) and show that the material is near fully-stressed and achieves high efficiency of pressure vessel (EPV). The authors state that a pressure vessel using the shellular material systems can be designed to any shape, thus fuel can be stored to conform to a vehicle component (giving an example of a car chassis) and is safer due to the leak-to-break failure.

With the Abaqus analysis, the results are highly dependent on the discretization and it is customary to show the asymptotic mesh analysis. Without this, it is unclear whether a valid mesh is used for the analyses. For the results to be reproducible, all numerical and material parameters should be given, including the discretization details and how the geometries are generated. Insufficient details are given in the Supplementary Information document. Since the errors are a function of the discretization, the finite element model results between different structures are often not directly comparable and this needs to be considered in the discussion. One major concern is that the material systems in this work is very thin, $0.001 < t/D < 0.01$. The element type S4 is more like the Kirchhoff type which is usually not applicable to $t/D < 0.01$ so the numerical results are questionable.

The advantages of the conformability and the leak-to-break failure mechanism are also unconvincing. The example of using the shellular pressure vessel also as a car chassis seems extremely challenging as there are a variety of mechanical loads that the chassis will be required to carry. (The majority of the chassis loading would be in bending and shear, which would be detrimental). In fact, many of the automotive and other vehicle components are subjected to mechanical and/or thermal loads and using these shellular structures filled with fuel (e.g. liquid hydrogen) would not be viable. The reason a typical pressure vessel is prone to the "catastrophic explosive failure" is because the stress in the material is near uniform and shell bending/buckling is inherently unstable, both are also characteristics of a shellular pressure vessel. Therefore, it is unclear whether the proposed system would indeed be useful as a fuel storage nor safe.

Despite the lack of realistic application of the shellular pressure vessel system, it may be an interesting scientific discovery. However, the TPMS materials are well-known and the authors have already published a number of papers on their shellular material. The work presented in this manuscript contains finite element analyses and the subsequent cold-stretching, which are reasonably standard in engineering of a pressure vessel. Thus, it is unclear to me what the novel scientific contribution is.

Replies to the reviewer's comments.

-Reviewer #1

The submitted manuscript deals with the novel design of pressure vessels called by the authors a shellular. In the 'Experimental results' section they describe the P-shellular type, although in previous sections there were described different types. Moreover, similar study was published by the authors in: Failure of P-surfaced Shellular subjected to internal pressure (<https://doi.org/10.1063/1.5066578>). **I consider this as the biggest weakness of the manuscript and it should be explained in the first place, what is the main novelty comparing to that work.** Moreover, the following issues should be addressed in the major revision:

In our previous work [1], we measured the critical pressures of P-surfaced shellulars with $3 \times 3 \times 2$ cells and composed of Ni-P, Cu, and silica, subjected to internal pressure. Although the specimens had substantial geometrical imperfections and were composed of shells with thicknesses of the order of a micrometer, the resistances of the Cu shellulars under internal pressure were close to that of a conventional cylindrical pressure vessel. However, for practical use as a pressure vessel, internal volume per total vessel weight is another important factor in addition to pressure resistance, and it should be adequately high. In this circumstance, we calculated the values of internal volume per total vessel weight of the Cu shellulars used in the referred paper and compared them to those of cylindrical pressure vessels. The values were revealed to be approximately only half of those of the cylindrical pressure vessels. In this paper, through a comprehensive study of the effects of parameters, such as the TPMS type, cell size, sealing caps, and sub-volumes, we succeeded in finding a route to achieve a novel pressure vessel that outperforms the conventional spherical vessels, even in terms of internal volume per total vessel weight. These findings are the main novelties of this paper over the previous paper. To address the reviewer's comment, we have included information pertaining to the calculation of internal volume per total vessel weight in the Supplementary Information, as follows:

“Internal volumes per total weight of previous shellular specimens

The following figures depict the three-dimensional geometry of the Ni-P shellular specimens, which were measured using a micro-CT¹. This geometry was similar to those of the specimens used in the internal pressure experiments² except of the sealing caps, which were inevitably needed to confine compressed fluid in the interior space. Thus, Kolesnikova et al.² also used the geometry to build their FEA models for stress analysis.

From these digital images, the surface area and internal volume of the specimens were calculated. By using these calculated values along with the shell thickness data provided in their paper, the internal volumes per total weight of each specimen were determined, as listed in Table S1.

Table S1. Internal volumes per total weight and critical pressure of the Cu shellular specimens tested by Kolesnikova et al.² and theoretical values of the corresponding cylindrical pressure vessel.

Shell thickness, t (mm)	Relative thickness, t/D	*Surface area, A (mm ²)	*Internal volume, V_{in} (mm ³)	**Solid volume, V_s (mm ³)	**Weight, m (g)	Internal volume per total weight (mm ³ /g)	***Internal volume per total weight (mm ³ /g)	Critical pressure, P_o (MPa)	***Critical pressure, P_o (MPa)
0.005	0.0025	211.104	45.648	1.056	0.009415	4848	9300	0.1	0.525
0.007	0.0035			1.478	0.01318	3463	6643	0.31	0.735
0.01	0.005			2.111	0.01883	2424	4650	0.65	1.05
								0.7	
								0.77	
0.012	0.006			2.533	0.02260	2020	3875	0.79	1.26
								0.87	
								0.94	

* calculated from the digital topology measured using a micro-CT

** calculated from the surface area and the thickness

*** theoretically estimated for the corresponding cylindrical pressure vessel as follows:

The internal volumes per total weight is given by

$$\frac{\left(\frac{\pi}{4}D^2l + \frac{\pi}{6}D^3\right)}{\rho t(\pi Dl + \pi D^2)} = \frac{1}{12} \frac{D(3l+2D)}{\rho t(l+D)} = \frac{1}{12\rho} \left(\frac{t}{D}\right)^{-1} \left[3 - \frac{D}{l+D}\right].$$

Here, ρ denotes the constituent material's density, and the conventional cylindrical pressure vessel is assumed to have a straight section of length, l , in the middle and hemispherical sealing caps on both sides. The yield pressures for a cylindrical pressure vessel is given by

$$P_o = 2\sigma_o \times \frac{t}{D}."$$

Additionally, two new paragraphs have been added at the end of the Introduction section of

the current manuscript as follows.

“Our proposal is based on the results of an experiment conducted by Kolesnikova et al.¹⁸ in 2019. They measured the critical pressures of *P*-surfaced shellulars with $3 \times 3 \times 2$ cells, composed of Ni-P, Cu, and silica, under internal pressure. The critical pressures of the most ductile shellular specimens, composed of Cu, were the highest among them, despite the low strength of Cu. Although the specimens exhibited substantial geometrical imperfections and were made of shells with thicknesses of the order of a micrometer, under internal pressure, the resistances of the Cu shellular specimens were close to that of a conventional cylindrical pressure vessel for a given t/D (Please note that the D values of shellular specimens indicated their unit cell sizes, whereas the D value of a conventional cylindrical pressure vessel indicates the diameter of its overall shape, and t indicated shell (or wall) thickness.).

Nevertheless, based on this result, one should not conclude that a shellular can be used as a pressure vessel before verifying whether its internal volume per total vessel weight is sufficiently high. For example, a truss-like structure composed of tubes and a matrix of spheres connected by tubes, as illustrated in Figs.1(b) and (c), respectively, cannot be used as pressure vessels, because their internal volumes are limited compared to their total weights, even if they exhibit high strength under pressure. In fact, the internal volumes per total weight of the Cu shellular specimens tested by Kolesnikova et al.¹⁸ were approximately only half those of the cylindrical pressure vessels with identical t/D values, as elaborated in the first section of **Supplementary Information**. Then, can a shellular not be a good pressure vessel? To answer this question, over the past four years, we have comprehensively studied the effects of parameters such as TPMS type, cell size, sealing caps, and sub-volumes before finally finding a route to achieve a novel pressure vessel that outperforms conventional pressure vessels. This article summarizes the findings of the aforementioned comprehensive investigation.”

[Reference]

[1] Kolesnikova, T., Wu, C. H., Han, S. C. & Kang, K. Failure of *P*-surfaced shellular subjected to internal pressure. *AIP Adv.* **9**, 025010; 10.1063/1.5066578 (2019).

1. Did you perform the analysis of mesh size element influence on the results?

Because FEA was used as the main method to elucidate the core concept of shellular pressure vessels herein, a preliminary study was conducted to investigate the effects of mesh size of finite elements, a key factor affecting the reliability of FEA results. For the cold-stretched shellular models with $3 \times 3 \times 3$ cells, which were used most frequently in this study, four different mesh sizes (0.5, 0.1, 0.07, and 0.02 mm) were employed to evaluate internal pressure resistance. The results indicated that for mesh sizes smaller than $70 \mu\text{m}$, the internal pressure performance and stress distribution remained almost constant, despite the additional reduction in mesh size. Therefore, herein, the mesh size of $70 \mu\text{m}$ was utilized to achieve adequately reliable results with minimal time investment. To address the reviewer's comment, we have added a new subsection in the Supplementary Information, as follows:

“Optimization of mesh size and element type

To secure the best accuracy with the minimal FEA run time, we investigated the effects of mesh size by using a double-chambered shellular vessel model composed of $3 \times 3 \times 3$ cells. Figs. S1a–d depict the Mises stress distributions of the four models with mesh sizes of 0.5, 0.1, 0.07, and 0.02 mm, respectively. Fig. S1e summarizes the variation in their relative yield pressures as a function of mesh size. According to figure, the mesh size of 0.07 mm ($70 \mu\text{m}$) provided the best accuracy with the minimal run time. Hence, we used this mesh size for most models.

Four-node shell elements (S4 of Abaqus®), which are known to provide robust and accurate solutions in all loading conditions for thick and thin shell problems were employed in the entire range of $t/D = 0.01$ to 0.0001 ^{4,5}. Their accuracy was validated by comparison with analytic solutions for conventional cylindrical and spherical pressure vessel models³.”

Fig. S1. Effects of the mesh size on FEA estimation. Von Mises stress distribution in the four models with the mesh size of a, 0.5, b, 0.1, c, 0.07, and d, 0.02 mm. e, comparison of relative yield pressures in four models. The yield pressures were estimated from FEA for cold-stretched double chambered models with an identical shell thickness of $t = 0.001D$.

2. What was the level of plastic deformation occurring in the material after cold-stretching?

As explained in the main text, a cold-stretched model was realized by applying pressure to the internal volume until more than 85% of the surface area yielded. The equivalent plastic strain generated within the model was monitored. As illustrated in the following figure, the equivalent plastic strain peaked at 8.6% in case of the double-chambered shellular vessel.

This information has been added to the main texts as follows:

“To obtain the cold-stretched models by conducting FEA, we applied excessive internal pressure to each model until more than 85% of its surface area underwent plastic yielding. Consequently, the maximum equivalent plastic strain on the surface reached 8.6%.”

3. How did you perform the strain measurements? Did you use Digital image correlation (DIC) method or just measuring the pixel displacements form regular photos?

In the tensile tests, displacements were measured by counting pixels in the digital images. To address the reviewer’s comment, we examined the accuracy of the pixel-counting method used herein by comparing its result for a specimen to the value measured using the digital image correlation (DIC) method. The comparison results are presented in the following figure. The strains measured using the pixel-counting method agreed fairly well with those measured using the DIC method, and the maximum error was 6%. Therefore, the values obtained using the pixel-counting method cannot be considered precise in the general engineering sense. The errors seemed to lead to variance in the slope and, consequently, in the Young’s modulus. However, because the tensile tests were conducted not to measure the Young’s modulus but to measure the yields strengths and elongations corresponding to the thickness of the plated Cu layer, the errors in the Young’s modulus were ignored in this study.

4. How would you explain the different Young Modulus for each type of foil? (as it can be observed in Fig. S7B). It may be noticed that the highest stiffness was obtained for 27.5 μm foils? Do you have explanation for that?

In response to the reviewer's comment, we conducted displacement measurements at five points in each specimen by using the pixel-counting method and re-plotted the average stress-strain curve. Accordingly, the stress-strain curves in the supplement information have been replaced with the modified ones as follows. Although the individual data related to elongation changed substantially, the upper and lower bounds remained unchanged. As mentioned above, because the tensile tests were conducted to measure the yield strengths and elongations according to the thickness of plated Cu layer, the errors in measurement of Young's modulus were ignored.

5. What type of machine was used in the mechanical tests with what range of force measurements? Was it proper for such a 'delicate' specimens?

We believe that the “mechanical test” mentioned by the reviewer means “internal pressure test”. To accurately measure the pressure resistance of these delicate specimens, the shell thicknesses of which are of the order of a few micrometers, we used a lower-capacity load cell in addition to the one built into the INSTRON mechanical test system. In response to the reviewer's comments, we have added two paragraphs and one photograph to the “Measurement of Yield Pressures of Shellulars” section in the Supplementary Information to provide detailed information about the test method and setup, as follows:

“The internal cavity of each specimen was filled with glycerol aqueous solution, and the specimen was attached to a syringe filled with the identical solution. The syringe attached to the specimen was then mounted on top of a rectangular frame, as shown in Figs. 3a and b, for the internal pressure test. The frame was mounted on an electro-hydraulic material test system, INSTRON 8872. Figure S8 depicts the test system used to measure the yield pressures of the shellular specimens. To apply such high pressures that the specimens with thick shells such as $t/D \approx 0.01$ yield, a stainless steel syringe (VCDS10, VMATIC Co., China) with a high-pressure male Luer Lock connector 80353 (Qosina Corp., NY, USA) adhesively bonded to a nozzle using Loctite® Epoxy Instant Mix™ (Henkel Corp., USA) was used. A needle (20 G, PrecisionGlide™, Becton-Dickinson & Co., USA) whose end was inserted into a specimen through PDMS sealing was attached to the Luer Lock connector.

The pressure was applied by pushing the syringe's plunger at a constant displacement rate of 0.01 mm/s. The plunger's displacement was measured using a linear variable displacement transducer (LVDT) built into the test system. In addition to the built-in load cell, a small-capacity load cell (CWFS-20, Bongshin Loadcell Co., Ltd., Republic of Korea) with a load capacity of 200 N was placed under the frame to measure the applied load more precisely. In cases of the specimens with t/D values higher than 0.00509, whose estimated loads exceeded 200 N, another small-capacity load cell (CWFS-100, Bongshin Loadcell Co., Ltd., Republic of Korea) with a load capacity of 1 kN was used for load measurement. The load and displacement data were acquired using a strain amplifier (2311 Signal Conditioning Amplifier, Vishay

Measurements Corp., USA) and data acquisition board (DT322 Data Translation, Spectrum Instrumentation Corp., USA), and stored in a personal computer. The load data were divided by the cross-sectional area of the syringe's plunger, the diameter of which was 14.8 mm, to convert them to the internal pressure applied to each shellular vessel specimen through the needle."

Fig. S8. Test setup for measuring yield pressures of shellular pressure vessel specimens.

6. The abstract should include some quantitative results regarding your research.

In response to the reviewer's comment, we have included a representative result related to internal volume-per-total weight, as follows:

“Nevertheless, for a given constituent material and prescribed pressure, the achievable internal volume-per-total weight of a double-chambered shellular vessel with cells of more than 15×15 can exceed the practical upper bound of both spherical and cylindrical vessels.”

-Reviewer 2

The paper is interesting and presents a promising technology that may improve pressure vessel technology the following comments may improve clarity of presentation and understanding

the paper deals with triply periodic minimal surfaces (TPMS). What does this mean? why are they called that?

A triply periodic minimal surface is a smooth surface with a constant mean curvature. Here, a minimal surface refers to a surface that is locally area-minimizing. The mean curvature used here corresponds to the average of the sectional curvatures rather than their sum. According to Han & Che [1], TPMSs are defined as follows:

“Triply periodic minimal surfaces (TPMSs) are composed of infinite, non-self-intersecting, periodic surface structures in three principal directions and are associated with the crystallographic space group symmetry [2]. The first examples of TPMSs were discovered by Schwarz in 1865 [3], followed by his student Neovius in 1883 [4]. They described five TPMSs, namely, Schwarz primitive (P), Schwarz diamond (D), Schwarz hexagonal (H), Schwarz crossed layers of parallels, and Neovius (N). Then, the most famous gyroid (G) surface was described by Schoen in 1970, along with another eleven newly discovered TPMSs [5].”

The following figures depict the unit cell and multiple cells of P-, D-, and G-surfaces as typical examples of TPMSs.

For more details, please visit a famous and reliable site being operated by Ken Bakke.

<https://kenbrakke.com/evolver/examples/periodic/periodic.html>

Hence, if a shellular has the shape of a TPMS, stress concentration does not occur in it because of no sudden changes in geometry, and local bending is suppressed under external loads. Instead, the shellular resists external loads only by coplanar stresses. Consequently, buckling (which is more likely to occur when the wall thickness decreases) is effectively delayed. Thus, a TPMS shellular can be considered to possess only the stretching-dominated architecture at ultralow densities [6]. The following figure presents an in-situ micro-CT image of a shellular structure with a TPMS (P-surface) shape under compression, revealing a uniform distribution of wrinkles [7]. This indicates that the shellular structure adopts a topology resembling that of the TPMS, which allows for uniform stress distribution. In a previous study, we confirmed that this leads to stretching-dominated deformation.

[References]

- [1] L. Han, S. Che, An Overview of materials with triply periodic minimal surfaces and related geometry: From biological structures to self-assembled systems. *Adv. Mater.* (2018) 30, 1705708.
- [2] S. Hyde, Z. Blum, T. Landh, S. Lidin, B. W. Ninham, *The Language of Shape: The Role of Curvature in Condensed Matter: Physics, Chemistry and Biology*, Elsevier Science B.V., Amsterdam, The Netherlands 1996.
- [3] H. A. Schwarz, *Gesammelte Mathematische Abhandlungen*, Springer, Berlin, Germany 1890.
- [4] E. R. Neovius, *Bestimmung Zweier Spezieller Periodischer Minimalflächen*, Akad. Abhandlungen, Helsinki, Finland 1883.
- [5] A. H. Schoen, *Infinite Periodic Minimal Surfaces Without Self-Intersections*, NASA Technical

Report D-5541, NASA, USA 1970.

- [6] V.S. Deshpande, M.F. Ashby, N.A. Fleck, Foam topology: bending versus stretching dominated architectures. *Acta Mater.* 49 (2001) 1035–1040.
- [7] S.C. Han, K. Kang, Another stretching-dominated architecture, shellular. *Mater. Today* (2019) 31, 31-38.

Equation 1. Parameter f was not defined

In response to the reviewer's comment, the parameter f has been defined as follows:

“And f denotes the volume fraction, defined as the ratio of one sub-volume to the overall volume. In our TPMS shellular pressure vessel design, we used a constant volume fraction of $f = 0.5$, meaning that the two sub-volumes were identical in a unit cell.”

table s1 shows cell sizes, shell thicknesses, surface areas, weights, and EPVs. This seems like important information. Should this be included in the main article?

It is true that Table S1 contains all the information, and it is important. However, the main purpose of the table is to compare the characteristics of conventional pressure vessels with those of single- and double-chambered shellular pressure vessels as a function of the number of cells when they have the same internal volume as that of sphere with a diameter of 1 m and are subjected to a constant yield pressure of $P_o = 0.01 \sigma_o$. These details demonstrate the variations in each factor and how they correspond to changes in the *EPV*. The conclusive changes in the *EPV* of each structure are more vividly depicted in Fig. 2d in the main text, and therefore, this information has been retained in the Supplementary Information for reference.

line 167 claims that the vessel efficiency is higher than that of a spherical vessel. Is this possible? As I remember, a sphere has a uniform stress distribution where every location has equal stresses in both directions along the surface. How can anything be better

than that?

First, it is important to note that shellular pressure vessels do not inherently possess superior internal pressure performance compared to those of traditional pressure vessels. At best, they perform at a level similar to that of cylindrical pressure vessels. Therefore, vessel efficiency superior to that of spherical pressure vessels can be achieved only under certain conditions. According to these conditions, superior vessel efficiency is achievable only with double-chambered shellular pressure vessels, not single-chambered ones. Additionally, a substantial number of cells, at least $15 \times 15 \times 15$ or more, should be used to construct a shellular pressure vessel that satisfies these criteria. The structure of a double-chambered shellular pressure vessel, illustrated in the newly added Fig. 4a in the revised manuscript, can be divided into two main components: an outer shell designed to withstand internal pressure and an interior frame designed to suppress deformation from within. Fig. 4e clarifies that when the internal volume and internal pressure are fixed, the overall size of the double-chambered shellular pressure vessel decreases slowly as the number of cells increases, as opposed to spherical pressure vessels. As a result, as depicted in Fig. 4b, for a given internal pressure, the vessel thickness decreases sharply as the number of cells increases.

Because of this phenomenon, the solid volume of the outer shell, which maintains a similar area regardless of the number of cells, decreases significantly. By contrast, the surface area of the interior frame increases consistently, which causes the solid volume to increase initially before converging to a constant value as the number of cells increases. In conclusion, when considering the entire structure, an increase in the number of cells leads to a sharp decrease in solid volume, which eventually converges to a constant value. These findings were tested using the newly derived general solutions presented in the Supplementary Information. The solutions of surface area and internal volume were verified through comparisons with the values obtained from CAD models. Consequently, the *EPV* of shellular pressure vessels exhibited a unique behavior as the number of cells increased. This behavior was attributed to the structural characteristics of the outer shell and interior frame. When composed of a substantial number of cells, such as $15 \times 15 \times 15$ or more, despite their lower internal pressure performance, shellular pressure vessels surpass the efficiency of spherical pressure vessels.

To provide detailed systematic explanations, a new subsection entitled "Feasibility of

shellular pressure vessels being superior to conventional spherical vessels" and a new Fig. 4 have been added to the Discussion section. In addition, two relevant sections, "General Solutions for Solid Volumes" and "Validation of Eqs. (S6), (S7), (S11), and (S12)" have been added to the Supplementary Information. The added sections are skipped here because they are lengthy. Please refer to the revised manuscript and Supplementary Information.

In a vessel like this made of individual components I believe sealing all the joints may be a challenging problem. How is this handled for a "real" vessel? if the individual p-surfaces need to be joined to individual outer membranes, how do I make sure that the joints will not leak? This should be explained in the main text.

As explained previously, double-chambered shellular pressure vessels can be divided into two parts: outer shell and interior frame. Because the actual pressure-bearing portion is confined to the outer shell, it is expected that the interior frame may be more insensitive to defects. This expectation aligns with the results of the experiments in which the yield pressures of the shellular vessel specimens that underwent cold stretching were measured. The single-chambered specimens exhibited a success rate of 19%, while the double-chambered specimens exhibited a higher success rate of 25%. To further investigate these aspects, an additional FEA of the cold-stretched double-chambered pressure vessel models was performed. First, a model composed of 3 x 3 x 3 cells was created with random defects in the interior frame, which accounted for 5.3% of the frame's total area. The internal pressure resistance and stress distribution of this model were compared to those of an intact model. The results of these comparisons, as shown in the newly added Fig. 5b, revealed that the stress distributions in the outer shell remained unchanged. Consequently, the internal pressure performance remained consistent despite the high stresses distributed within the interior frame. As mentioned previously, this confirmed the insensitivity of the interior frame in double-chambered pressure vessels to defects. Furthermore, FEA was conducted on a double-chambered vessel with a greater number of cells, configured as 9 x 9 x 9, by removing cells from the interior frame. When one central cell was removed, as depicted in the newly added Fig. 5a, the stress distribution in the outer shell and the internal pressure performance remained the same as those of the intact model. However, when a 3 x 3 x 3 set of central cells

was removed, the absence of cells inside led to noticeable stress concentration in the outer shell, leading to changes in stress distribution and degradation of internal pressure performance. This result revealed that the interior frame played an important role to support the outer shell and, consequently, to preserve the entire structure against internal pressure.

To explain this issue in detail, a new subsection "Technical difficulty associated with its fabrication" and a new Fig. 5 have been added to the Discussion section. The added section is skipped here for simplicity. Please refer to the revised manuscript.

To facilitate the production of sufficiently large pressure vessels for use in practical applications, we introduce a method involving the assembly and welding of quadrilaterals. TPMS shellular structures can be fabricated by combining many constant quadrilaterals along an anti-clastic curvature. This can be achieved by assembling thin sheets in the shape of anti-clastic curvature squares and, subsequently, connecting them using welding methods such as laser welding. As explained previously, the interior frame of a double-chambered pressure vessel allows for some degree of defects, thus making it advantageous for practical fabrication. The following paragraph has been added to explain this.

“Because a TPMS is composed of many constant quadrilaterals in an anti-clastic curvature²⁴, thin shells are cut and bent into a regular quadrilateral shape and then welded to each other to build a large-scale shellular pressure vessel, as illustrated in Fig. 5c. In this case, although the thin shells themselves are easy to weld to each other by using a focused energy source, such as a fiber laser²⁵, the process of welding along the complicated contours in three-dimensional space would be technically challenging. Nevertheless, because defects are allowed in the interior frame, as mentioned above, it should be feasible to manufacture large-scale shellulars by employing a robot-based rough technology to build the interior frame, while employing a more precise technology to guarantee defect-free welding for building the outer shell, as in the fabrication of a conventional pressure vessel.”

Additionally, all shellular pressure vessels undergo shape optimization through cold stretching. In this process, pressure is applied to achieve the desired shape. Consequently, it is possible to detect defects in the outer shell of pressure vessels during this cold stretching process.

line 203 mentions equations 6 and 7 but these are not included in the text

In response to the reviewer's comment, the errors have been corrected as follows:

“For comparison, the estimations by elementary mechanics (Eqs. (S2) and (S3) in Supplementary Information) for conventional pressure vessels are plotted as the solid black and dashed grey lines, respectively.”

the authors mention hydrogen storage as a potential application. However, I question whether a metal-based system like this (even when covered with graphene) would be excessively heavy for automotive applications?

It is true that Type 4 pressure vessels, which are widely used in automotive applications, are considerably lighter (70–75%) than Type 1 pressure vessels made solely of metal, thanks to the use of wound composite materials [1]. However, large-scale composite pressure vessels are not likely to be cost-effective anymore. Indeed, hydrogen pressure vessels composed of a metal structure are being used not only as fuel tanks but also in various other applications. The first example is multiple-element gas containers (MEGC), which are essentially assemblies of conventional cylindrical tanks for transporting large volumes of hydrogen. These classical high-pressure tanks, typically made of cost-effective steel, are tested for pressures of up to 300 bar and are regularly filled up to 200 bar in most countries [2]. MEGCs serve as mobile or stationary devices for the transportation, storage, and distribution of hydrogen. They are suitable for both inland and international transportation, including road and rail, and they eliminate the need to transfer the medium to another vessel. The structure of MEGCs is realized by mounting pressure vessels within a shipping container frame, as depicted in the following diagram. Owing to this design, in a MEGC container with 12 ISO-standard tanks, each measuring 36 feet in length, the hydrogen-carrying cylindrical pressure vessels account for 88.2% of the total container weight, and most of this is the weight of the pressure vessels

themselves [3]. By replacing these cylindrical pressure vessels with shellular pressure vessels, according to our efficiency calculations, we expect that the weight of the pressure vessels required to transport the same volume of hydrogen can be reduced by up to 46.8%, which would significantly reduce the transportation cost.

The second example of the use of a metal structure to store hydrogen is a stationary large-capacity cryogenic hydrogen storage tank. Hydrogen is widely stored in liquid form in large volumes because the density of saturated liquid hydrogen at 1 bar is 70 kg/m³ [4] while that of gaseous hydrogen compressed to 700 bar is only 42 kg/m³. Liquid hydrogen has mainly been evaluated as a hydrogen distribution medium, and high density is a substantial advantage in this regard [5]. However, the liquefaction of hydrogen requires significant energy input owing to its extremely low boiling point (-253 °C at 1 bar) [6]. Furthermore, after hydrogen has been liquefied, it is essential to store it properly to minimize its evaporation. The evaporation of liquid hydrogen amounts to a loss of not only the energy spent for liquefaction but also a loss of hydrogen because the evaporated gas must be vented to prevent pressure build-up inside the storage vessel. This loss of stored hydrogen over time is called boil-off, and it is often presented as a percentage of stored hydrogen lost per day, that is, the boil-off rate. Therefore, liquid hydrogen storage vessels are most commonly double-walled, and a strong vacuum is applied between the walls [6]. This is because vacuum insulation minimizes heat transfer through conduction and convection [7]. As a prime example of large-scale liquid hydrogen storage, the world's largest liquid hydrogen storage tanks were built in the mid-1960s at NASA's Kennedy Space Center [8,9]. These liquid hydrogen vessels have a diameter of 21 m and can store 850,000 gallons of liquid hydrogen. They are spherical and incorporate a thick layer of vacuum-jacketed insulation filled with perlite. This design allowed NASA to achieve a normal boil-off rate as low as 0.0625%.

Replacing liquid hydrogen storage containers with shellular pressure vessels offers several advantages. First, shellular pressure vessels surpass spherical tanks in terms of *EPV* (internal volume per total vessel weight for a given pressure and constituent material) while eliminating the need to incorporate high-vacuum double walls for cryogenic storage, which increases the storage capacity of vessels. Additionally, these vessels are composed of a periodic combination of numerous unit cells. Therefore, even if the overall vessel size increases, the thickness of the unit cells and shell can be maintained by simply increasing the number of

cells. This means that unlike spherical containers, no size constraints are imposed by wall thickness. Therefore, despite storing relatively low-density gaseous hydrogen, shellular pressure vessels are expected to remain highly competitive for large-scale storage. Furthermore, they can contribute to cost-effectiveness by reducing the energy consumption associated with the liquefaction of hydrogen, maintaining high vacuum, and re-liquefaction of boiled-off gas, which occur in liquid hydrogen storage.

Simple schematics of MEGC

World's largest liquid hydrogen storage tanks in NASA

[References]

- [1] Azeem. M. et al. Application of Filament Winding Technology in Composite Pressure Vessels and Challenges: A Review, *J. Energy Storage*. **49**, ,103468;

- 10.1016/j.est.2021.103468 (2022).
- [2] Schlapbach, L., Züttel, A., Hydrogen-storage materials for mobile applications. *Nature* **414**, 353–358 (2001).
- [3] MEGC 40 Ft - 12 Tubes UN ISO 11120 E & NE Gas 2654 Psi 36 Ft, Available at: <http://cmwelding.com/megc-40-ft-12-tubes-un-iso-11120-e-ne-gas-2654-psi-36-ft> (Accessed: 20th September 2023)
- [4] Godula-Jopek, A., Jehle, W., Wellnitz, J., *Storage of pure hydrogen in different states. In: Hydrogen storage technologies*. 97-170 (WileyVCH Verlag GmbH & Co. KGaA, 2012).
- [5] Cardella, U., Decker, L., Klein, H., Roadmap to economically viable hydrogen liquefaction. *Int J Hydrogen Energy* **42** (19), 13329-13338 (2017).
- [6] Andersson, J., Grönkvist, S., Large-scale storage of hydrogen, *Int. J. Hydrog. Energy* **44** (23), 11901-11919 (2019).
- [7] Klell, M., *Handbook of hydrogen storage: new materials for future energy storage*. 1 (Wiley-VCH Verlag GmbH & Co. KGaA, 2010).
- [8] NASA Press Release, Liquid Hydrogen--the Fuel of Choice for Space Exploration, Available at: <https://www.nasa.gov/content/liquid-hydrogen-the-fuel-of-choice-for-space-exploration> (Accessed: 20th September 2023)
- [9] Fesmire, J., Swanger, A., Jacobson, J., Notardonato, W., Energy efficient large-scale storage of liquid hydrogen, *IOP Conf. Series: Materials Science and Engineering* **1240** 012088,10.1088/1757-899X/1240/1/012088 (2022).

Lines 246-247: the authors report "tensile strength" for graphene, "fracture strength" for boron nitride, and "strength" for transition metal dichalcogenides. Do these three terms refer to the same thing? If they do, the authors should use the same term (tensile strength?) for all. Otherwise, why do they report different parameters for the three?

The strengths of the 2D materials were not measured by conducting typical tensile tests. Therefore, the measured values are not tensile strengths. Actually, all the strength values were measured by nano-indentation of free-standing membranes in an atomic force microscope,

and they were termed "breaking strength." Thus, as suggested by the reviewer, the terminology has been standardized as "breaking strength" in the revised manuscript.

“Therefore, the surface is favorable for conformal deposition of an ultra-strong 2D material, such as graphene with a breaking strength of 130.5 GPa²⁶, boron nitride (BN) with breaking strength of 70.5GPa²⁷, and transition metal dichalcogenides (MoS₂ with a breaking strength of 27 GPa)^{28,29}.”

-Reviewer 3

This paper presents a numerical and experimental study of shellular materials as a pressure vessel. The authors have published a number of papers on shellular material systems, as a form of TPMS which have been studied for a wide range of applications and also found in many biological systems. This study employs the finite element method (using Abaqus) to conduct the stress analyses, apply some plastic deformation based on the yield stress of the material (cold-stretched) and show that the material is near fully-stressed and achieves high efficiency of pressure vessel (EPV). The authors state that a pressure vessel using the shellular material systems can be designed to any shape, thus fuel can be stored to conform to a vehicle component (giving an example of a car chassis) and is safer due to the leak-to-break failure.

With the Abaqus analysis, the results are highly dependent on the discretization and it is customary to show the asymptotic mesh analysis. Without this, it is unclear whether a valid mesh is used for the analyses. For the results to be reproducible, all numerical and material parameters should be given, including the discretization details and how the geometries are generated. Insufficient details are given in the Supplementary Information document. Since the errors are a function of the discretization, the finite element model results between different structures are often not directly comparable and this needs to be considered in the discussion. One major concern is that the material systems in this work is very thin, $0.001 < t/D < 0.01$. The element type S4 is more like the Kirchhoff type which is usually not applicable to $t/D < 0.01$ so the numerical results are questionable.

In response to the reviewer's comment, we have added a new subsection in the supplementary Information to explain the details of FEA procedure, where we specifically address the discretization-related concerns raised by the reviewer as follows:

“Optimization of mesh size and element type

To secure the best accuracy with the minimal FEA run time, we investigated the effects of mesh size by using a double-chambered shellular vessel model composed of $3 \times 3 \times 3$ cells. **Figs. S1a–d** depict the Mises stress distributions of the four models with mesh sizes of 0.5, 0.1, 0.07, and 0.02 mm, respectively. **Fig. S1e** summarizes the variation in their relative yield pressures as a function of mesh size. According to figure, the mesh size of 0.07 mm (70 μm) provided the best accuracy with the minimal run time. Hence, we used this mesh size for most models.

Four-node shell elements (S4 of Abaqus®), which are known to provide robust and accurate solutions under all loading conditions in thick and thin shell problems, were employed in the entire t/D range of 0.01–0.0001^{4,5}. Their accuracy was validated by comparison with the analytical solutions of the conventional cylindrical and spherical pressure vessel models³.”

Fig. S1. Effects of mesh size on FEA results. Von Mises stress distribution in the four models of double-chambered shellular vessel with the mesh size of **a**, 0.5, **b**, 0.1, **c**, 0.07, and **d**, 0.02 mm. **e**, comparison of relative yield pressures in the four models of cold-stretched double-chambered shellular vessel with an identical shell thickness of $t = 0.001D$.”

Additionally, according to Abaqus manual, general-purpose shell elements, that is, S4 elements, provide robust and accurate solutions under all loading conditions for thick and thin

shell problems [1,2]. In the case of thin shell elements (where the thickness of the shell element is less than 1/15th the reference length), only convergence in the plastic region is problematic. Therefore, the element type can be applied regardless of the t/D ratio used in this research. The reviewer's concern that the S4 element is more like the Kirchhoff element, which is usually not applicable to $t/D < 0.01$, has already been addressed by Wu [3], who conducted FEA in the same way as that in our study and compared the results with those obtained using analytical solutions for conventional cylindrical and spherical pressure vessels. For a spherical pressure vessel, the critical pressures corresponding to plastic yielding [4] and elastic buckling [5] are expressed as (1) and (2), respectively:

$$P_{cr} = \frac{4t \times \sigma_o}{D} \quad \text{--- (1)}$$

$$P_{cr} = \frac{8E}{\sqrt{3(1-\nu^2)}} \times \left(\frac{t}{D}\right)^2 \quad \text{--- (2)}$$

For a cylindrical pressure vessel, the critical pressures corresponding to plastic yielding and elastic buckling [6] are expressed as (3) and (4), respectively:

$$P_{cr} = \frac{2t \times \sigma_o}{D} \quad \text{--- (3)}$$

$$P_{cr} = \frac{E}{1-\nu^2} \times \left(\frac{t}{D}\right)^{2.2} \quad \text{--- (4)}$$

The following figure compares the critical pressures of the conventional pressure vessels estimated by means of FEA with those calculated using the above analytical solutions.

The overall dimensions and element sizes of the finite-element models were similar to those of the unit cells of the TPMS shellular vessel models analyzed herein, and the element types (type S4) were the same as those of the shellular vessel models. The FEA estimates agreed fairly well with the analytical solutions, regardless of the pressure vessel type and failure mode, which validated the FEA procedure. Notably, the agreements were valid over the entire relative thickness range of $t/D = 10^{-5}$ to 10^{-2} , which addresses the reviewer’s concern.

[References]

- [1] Dassault Systèmes Simulia Corp (2006). Abaqus Analysis User’s Manual, Available at: <https://classes.engineering.wustl.edu/2009/spring/mase5513/abaqus/docs/v6.6/books/usb/default.htm?startat=pt06ch23s06alm15.html> (Accessed: 19th September 2023)
- [2] Dassault Systèmes Simulia Corp (2006). ABAQUS Theory Manual, Available at: <https://classes.engineering.wustl.edu/2009/spring/mase5513/abaqus/docs/v6.6/books/stm/default.htm?startat=ch03s06ath79.html> (Accessed: 19th September 2023)
- [3] Wu. C. H., Failure Study of Shellulars under Internal Pressure. Master thesis, Graduate School, Chonnam National University (2019).
- [4] Lardner. T. J., Archer R. R., *Mechanics of Solids: An Introduction*, 528. (McGraw-Hill Book Company, 1994).

- [5] Timoshenko. S. P., Gere. J. M., Theory of Elastic Stability, 289 (McGraw-Hill Book Company, 1961).
- [6] Glock. D., Post-Critical Behavior of a Rigidly Encased Circular Pipe Subject to External Water Pressure and Thermal Extension, *Stahlbau*. **7**, 212–217 (1977).

The advantages of the conformability and the leak-to-break failure mechanism are also unconvincing. The example of using the shellular pressure vessel also as a car chassis seems extremely challenging as there are a variety of mechanical loads that the chassis will be required to carry. (The majority of the chassis loading would be in bending and shear, which would be detrimental). In fact, many of the automotive and other vehicle components are subjected to mechanical and/or thermal loads and using these shellular structures filled with fuel (e.g. liquid hydrogen) would not be viable. The reason a typical pressure vessel is prone to the “catastrophic explosive failure” is because the stress in the material is near uniform and shell bending/buckling is inherently unstable, both are also characteristics of a shellular pressure vessel. Therefore, it is unclear whether the proposed system would indeed be useful as a fuel storage nor safe.

We agree that the car chassis was inappropriate as an example of the practical application of shellular pressure vessels because various modes of mechanical loads (such as bending and twisting) and thermal stresses are applied simultaneously to an actual car chassis, which functions mainly as a structural frame rather than a pressure vessel. Therefore, we have deleted the related texts and figure from the revised manuscript. The car chassis was considered as an example in the first place to demonstrate the conformability of the proposed shellular vessels. Following would be a better example for the demonstration. Nevertheless, the conformability of the shellular vessels is adequately clear, and it need not be explained using any specific example or figure Thus, we have revised the text to simply mentioning the conformability provided by the periodic micro-architecture of the TPMS.

Safety and reliability are important factors when developing any new technology or product. For pressure vessels, leak-before-break (LBB) is widely recognized as a critical methodology for securing structural safety. To use LBB as the fail-safe criterion, it must be demonstrated that any credible defect would grow through the wall in a stable manner and create a detectable leak [1]. That is, LBB can be achieved when a defect grows to be a through-wall crack, which lets compressed gas to leak and relieves high pressure before the crack propagates rapidly and unstably to cause a catastrophic burst [2]. In this regard, a thinner wall guarantees a higher probability of leakage. Considering that the wall thickness-to-cell size ratio of the shellular vessel specimens used in the actual internal pressure tests was $t/D = 0.022\text{--}0.00079$, the shellular vessels fall in the category of extremely thin pressure vessels, similar to aluminum beverage cans or propane gas tanks among conventional pressure vessels. Therefore, it is reasonable to consider that the shellular vessels satisfy the LBB principle.

[References]

- [1] Bourga, R., Moore, P., Janin, Y. J., Wang, B. & Sharples, J. Leak-before-break: global perspectives and procedures. *Int. J. Press. Vessels Pip.* **129-130**, 43-49 (2015).
- [2] Irwin G. *Fracture of pressure vessels. Materials for Missiles and Spacecraft.* 204 -229. (McGraw-Hill, 1963).

Despite the lack of realistic application of the shellular pressure vessel system, it may be an interesting scientific discovery. However, the TPMS materials are well-known and the authors have already published a number of papers on their shellular material.

The work presented in this manuscript contains finite element analyses and the subsequent cold-stretching, which are reasonably standard in engineering of a pressure vessel. Thus, it is unclear to me what the novel scientific contribution is.

It is true that cold-stretching is a reasonably standard process in the engineering of conventional pressure vessels. In this study, we used cold stretching to modify the local geometries of the shellular vessels. However, we believe that the proposed micro-architected pressure vessel has the potential to revolutionize the energy-related industry in the emerging era of green mobility by replacing conventional spherical or cylindrical vessels that have been the sole option for more than 300 years.

Since the first paper published in 2015, the authors' research group has published nine shellular-related papers to date [1-9]. Specifically, three papers pertain to optimization of the shellular structure [1-3], two papers describe the application of shellular as tissue engineering scaffolds [4-5], and three papers describe the fabrication process of shellulars and their mechanical properties under uniaxial compression [6-8]. The paper by Kolesnikova et al. [9] was the only one in which the internal pressure resistance of shellular or possibility of using it as a pressure vessel was investigated.

To explain how the present study differs from our previous study, i.e., Kolesnikova et al. [9], two new paragraphs have been added at the end of Introduction section as follows:

“Our proposal is based on the results of an experiment conducted by Kolesnikova et al.¹⁸ in 2019. They measured the critical pressures of P-surfaced shellulars with $3 \times 3 \times 2$ cells, composed of Ni-P, Cu, and silica, under internal pressure. The critical pressures of the most ductile shellular specimens, composed of Cu, were the highest among them, despite the low strength of Cu. Although the specimens exhibited substantial geometrical imperfections and were made of shells with thicknesses of the order of a micrometer, under internal pressure, the resistances of the Cu shellular specimens were close to that of a conventional cylindrical pressure vessel for a given t/D (Please note that the D values of shellular specimens indicated their unit cell sizes, whereas the D value of a conventional cylindrical pressure vessel indicates the diameter of its overall shape, and t indicated shell (or wall) thickness.).

Nevertheless, based on this result, one should not conclude that a shellular can be used as a

pressure vessel before verifying whether its internal volume per total vessel weight is sufficiently high. For example, a truss-like structure composed of tubes and a matrix of spheres connected by tubes, as illustrated in Figs.1(b) and (c), respectively, cannot be used as pressure vessels, because their internal volumes are limited compared to their total weights, even if they exhibit high strength under pressure.

In fact, the internal volumes per total weight of the Cu shellular specimens tested by Kolesnikova et al.¹⁸ were approximately only half those of the cylindrical pressure vessels with identical t/D values, as elaborated in the first section of Supplementary Information. Then, can a shellular not be a good pressure vessel? To answer this question, over the past four years, we have comprehensively studied the effects of parameters such as TPMS type, cell size, sealing caps, and sub-volumes before finally finding a route to achieve a novel pressure vessel that outperforms conventional pressure vessels. This article summarizes the findings of the aforementioned comprehensive investigation.”

In addition, to explain how a shellular vessel can outperform a spherical vessel in a systematic way contrarily to general belief, a new subsection “Feasibility of shellular pressure vessels being superior to conventional spherical vessels” with new Fig. 4 has been added in Discussion section. Also, two relevant sections, “General Solutions for Solid Volumes” and “Validation of Eqs. (S6), (S7), (S11), and (S12)” have been added in the Supplement Information. The added sections are skipped here because they are lengthy. Please see the revised manuscript and supplementary information.

[References]

[1] Lee. M. G., Lee. J. W., Han. S. C., Kang. K., Mechanical Analyses of “Shellular”, an Ultralow-

- density Cellular Metal. *Acta Mater.* **103**, 595-607 (2016).
- [2] Nguyen. B. D., Cho. J. S., Kang. K., Optimal Design of “Shellular”, a Micro-Architected Material with Ultralow Density. *Mater. Des.* **95**, 490-500 (2016).
- [3] Nguyen. B. D., Han. S. C., Jeong. Y. C., Kang. K., Design of the P-Surfaced Shellular, an Ultra-Low Density Material with Micro-Architecture, *Comput. Mater. Sci.* **139** 162–178 (2017).
- [4] Tan. S., Gu. J., Han. S. C., Lee. D., Kang. K., Design and fabrication of a non-clogging scaffold composed of semi-permeable membrane. *Mater. Des.* **142**, 229–239 (2018).
- [5] Gu. J., Jeong. Y., Na. J. Y., Seon. J. G., Lee. D., Kang. K., Application of Semi-Permeable Membrane for a Scaffold in a Nature-Mimicking Vascular System. *J. Membr. Sci.* **611**, 118384; 10.1016/j.memsci.2020.118384 (2020).
- [6] Han. S. C., Lee. J. W., Kang. K., A New Type of Low Density Material; Shellular. *Adv Mater.* **27**, 5506-5511, (2015).
- [7] Han. S. C., Choi. J. M., Liu. G., Kang. K., A Microscopic Shell Structure with Schwarz’s D-Surface. *Sci. Rep.* **7**, 13405; 10.1038/s41598-017-13618-3 (2017).
- [8] Han. S. C., Kang. K., Another Stretching-Dominated Micro-Architected Material, Shellular. *Mater Today.* **31**, 31-38 (2019).
- [9] Kolesnikova, T., Wu, C. H., Han, S. C. & Kang, K. Failure of P-surfaced shellular subjected to internal pressure. *AIP Adv.* **9**, 025010; 10.1063/1.5066578 (2019).

REVIEWERS' COMMENTS

Reviewer #1 (Remarks to the Author):

The authors referred successfully to the questions raised in the review. The manuscript is now satisfactorily improved including some new information and additional paragraphs. The proposed design seems to be very promising in the hydrogen storage industry. I recommend the manuscript to publication in the new, revised form which have been submitted.

Reviewer #2 (Remarks to the Author):

the authors have satisfactorily addressed my comments. The paper is acceptable with no further changes.